# Causal contribution and dynamical encoding in the striatum during evidence accumulation

Michael M Yartsev[1,2†]*, Timothy D Hanks[1,3,4†]*, Alice Misun Yoon[1], Carlos D Brody[1,5]*

[1]Princeton Neuroscience Institute, Princeton, United States; [2]Department of Bioengineering, Helen Wills Neuroscience Institute, Berkeley, United States; [3]Department of Neurology, University of California, Davis, Sacramento, United States; [4]Center for Neuroscience, University of California, Davis, Davis, United States; [5]Howard Hughes Medical Institute, Maryland, United States

**\*For correspondence:**
myartsev@berkeley.edu (MMY);
thanks@ucdavis.edu (TDH);
brody@princeton.edu (CDB)

†These authors contributed
equally to this work

**Competing interests:** The
authors declare that no
competing interests exist.

**Reviewing editor:** Joshua I
Gold, University of Pennsylvania,
United States

**Abstract** A broad range of decision-making processes involve gradual accumulation of evidence over time, but the neural circuits responsible for this computation are not yet established. Recent data indicate that cortical regions that are prominently associated with accumulating evidence, such as the posterior parietal cortex and the frontal orienting fields, may not be directly involved in this computation. Which, then, are the regions involved? Regions that are directly involved in evidence accumulation should directly influence the accumulation-based decision-making behavior, have a graded neural encoding of accumulated evidence and contribute throughout the accumulation process. Here, we investigated the role of the anterior dorsal striatum (ADS) in a rodent auditory evidence accumulation task using a combination of behavioral, pharmacological, optogenetic, electrophysiological and computational approaches. We find that the ADS is the first brain region known to satisfy the three criteria. Thus, the ADS may be the first identified node in the network responsible for evidence accumulation.
DOI: https://doi.org/10.7554/eLife.34929.001

## Introduction

All behaving animals must interpret sensory information arriving from the environment and use that information to select future actions. How the nervous system solves this problem has been a long-standing question in neuroscience. Multiple studies across a wide range of behavioral tasks and model systems, including humans (*Hunt et al., 2012*; *Krajbich et al., 2012*; *Ratcliff et al., 2015*), non-human primates (*Gold and Shadlen, 2007*; *Huk and Shadlen, 2005*; *Shadlen and Newsome, 1996*) and rodents (*Brunton et al., 2013*; *Carandini and Churchland, 2013*; *Erlich et al., 2015*; *Hanks et al., 2015*; *Raposo et al., 2012*; *Sanders and Kepecs, 2012*) have proposed a framework through which neural circuits gradually accumulate sensory evidence to guide decisions. Yet, despite the observation of neural correlates of evidence accumulation in several brain regions (*Ding and Gold, 2010*; *Gold and Shadlen, 2007*; *Hanks et al., 2015*; *Ratcliff et al., 2007*; *Shadlen and Newsome, 1996*), a major challenge in this line of research has been that the neural circuits that are causally responsible for evidence accumulation have not yet been determined. Two of the cortical regions that are most prominently associated with evidence accumulation, namely the posterior parietal cortex (LIP; *Huk and Shadlen, 2005*; *Kira et al., 2015*; *Roitman and Shadlen, 2002*; *Shadlen and Newsome, 1996*) and the frontal eye fields in primates (FEF; *Ding and Gold, 2012a*; *Gold and Shadlen, 2000*; *Mante et al., 2013*), together with its probable rodent analogue, the frontal orienting fields (FOF; *Erlich et al., 2011*), have been the focus of studies

recently. Surprisingly, these studies have indicated that neither region is central to the computation of gradually accumulating evidence (*Erlich et al., 2015*; *Hanks et al., 2015*; *Katz et al., 2016*).

The anterior dorsal striatum (ADS) serves as an intriguing alternative candidate (*Ding and Gold, 2013*), due in part to its unique anatomical positioning as a convergence hub for multiple brain regions (*Cheatwood et al., 2003*; *McGeorge and Faull, 1989*) where neural signatures of evidence accumulation have been observed (such as the PPC and FEF/FOF; *Gold and Shadlen, 2007*; *Cheatwood et al., 2003*; *Ding and Gold, 2013*; *McGeorge and Faull, 1989*). The ADS is thus ideally positioned to participate in evidence accumulation as part of its established role in action selection (*Bogacz and Gurney, 2007*; *Graybiel, 2008*; *Hikosaka et al., 2014*; *Jin and Costa, 2010*; *Nelson and Kreitzer, 2014*; *Redgrave et al., 2010*). Modeling work has also suggested that the ADS may participate in post-accumulation decision commitment (*Lo and Wang, 2006*).

The auditory input to a different striatal subregion, the posterior 'auditory' striatum, has been shown to be critical for auditory discriminations, leading to the suggestion that cortical projections into the striatum may provide a general mechanism for the control of motor decisions (*Xiong et al., 2015*; *Znamenskiy and Zador, 2013*). Specifically with regard to evidence accumulation, work in primates found neural correlates of evidence accumulation in the ADS (Ding 2015; *Ding and Gold, 2010*; *Ding and Gold, 2012b*; *Ding and Gold, 2013*; *Ding, 2015*, *Lo and Wang, 2006*), and revealed that electrical microstimulation of the ADS impacts behavior that is based on accumulation of evidence (*Ding and Gold, 2012a*). These data led to the proposal that the ADS may contribute to the computations specifically involved in evidence accumulation. Yet three critical questions to test this proposal have been left unanswered.

First, is the ADS required for unimpaired accumulation-based decision making? To date, there have been no recorded inactivations of the dorsal striatum during the accumulation of evidence. Inactivations are important probes of whether a region plays a central causal role for a cognitive variable of interest (*Newsome and Paré, 1988*; *Erlich et al., 2015*; *Katz et al., 2016*).

Second, do neurons in the dorsal striatum encode sensory information in a way that is sufficient to be involved directly in the graded accumulation process? The correlates of evidence accumulation reported to date in striatum have been of two types: either firing rates that, when averaged over trials, ramp upwards with a slope of the ramp that increases as the evidence strengthens (*Ding and Gold, 2010*), or estimates of the temporal dynamics of firing rate variance across trials (*Ding, 2015*). However, the trial averages do not distinguish between graded evidence encodings and other encodings that on a single-trial basis do not represent gradually accumulating evidence, such as sharp coordinated steps in firing for which the timing of the step varies across trials (*Hanks et al., 2015*; *Latimer et al., 2015*). Furthermore, the variance estimates have not yet produced clearly definitive conclusions, suggesting that the ADS is only partly involved in graded accumulation (*Ding, 2015*).

We recently developed a complementary approach, distinct from the two earlier methods, to assess evidence accumulation encoding. This most recent approach estimates 'tuning curves,' that is, direct descriptions of the relationship between recorded neural firing rates and the graded value of the evidence accumulator, and can discriminate between different encodings that otherwise appear indistinguishable (*Hanks et al., 2015*). Here, we apply this approach to electrophysiological recordings from the ADS.

Third, does the dorsal striatum play a causal role throughout the period of accumulation? One of the key aspects of interest in gradual evidence accumulation is its relatively long timescale, as it occurs over a period of hundreds of milliseconds or more (thought to be a potential model of mental deliberation [*Gold and Shadlen, 2007*]). If the striatum is part of the gradual accumulation process that drives behavior, perturbing it at any timepoint during that accumulation process should affect behavior. This feature is thus an essential prerequisite for a component of the accumulator. However, no temporally specific perturbations of the ADS during the accumulation of evidence have yet been carried out to probe for this feature. Indeed, no brain region studied during an accumulation of evidence behavior has yet been reported to possess this feature.

Here, using a combination of behavioral, pharmacological, optogenetic, electrophysiological and computational approaches, we address these three fundamental questions. The results provide evidence supporting a central causal role for the anterior dorsal striatum in evidence accumulation.

## Results

We trained rats on a previously developed decision-making task (*Brunton et al., 2013*) in which subjects accumulate auditory evidence over many hundreds of milliseconds to inform a binary left/right choice (*Figure 1a*). In each trial, rats kept their nose in a central port during the presentation of two simultaneous trains of randomly timed auditory clicks, one played from a speaker to their left and the other from a speaker to their right. At the end of the auditory stimulus, the rat's task was to decide which side had played the greater total number of clicks. Consistent with previous studies using this task, analysis of our rats' behavior indicated that they gradually accumulated auditory evidence over the entire trial, and used that accumulated evidence to drive a categorical choice (*Figure 1—figure supplement 1*; *Supplementary file 1*).

We began to assess the role of the anterior striatum in the accumulation task using reversible pharmacological inactivation with muscimol (Materials and methods). The anterior striatal region targeted in this study receives convergent inputs from the PPC and the FOF, brain regions previously reported to contain neural correlates of evidence accumulation but later shown to not be central to the accumulation process itself (*Erlich et al., 2015*; *Hanks et al., 2015*; *Katz et al., 2016*). Unilateral inactivation of the ADS biased rats to make more ipsilateral choices relative to controls (*Figure 1b*; bias for right side inactivation = 19.2 ± 4.4%, p<0.01; bias for left side inactivation = 18.6 ± 3.3%, p<0.01). This effect was not a gross motor bias, but was instead specific for accumulation trials, because no significant bias was caused on interleaved motor control trials in which the rats had to make a similar left/right motor response, but were cued by a simple visual stimulus (*Figure 1—figure supplement 2*; p>0.4 for both left- and right-side trials). Bilateral pharmacological inactivation caused a substantial impairment in performance for accumulation trials (*Figure 1c*; impairment = 12.6 ± 3.2%, p<0.01). This impairment was again specific for accumulation trials, with no significant impairment in motor control trials where the decision was not based on the accumulation of evidence over time (*Figure 1—figure supplement 2*; p>0.6 for both left- and right-side trials).

Psychometric curves such as those shown in *Figure 1b,c* group together trials based on the click difference accrued by the end of the stimulus stream and treat all trials within each group as if they were the same. But in our clicks task, we have far more information available because the precise temporal pattern of each individual trial's click trains is known. We have previously used this information, together with a model that takes into account those known individual click times, to quantify our subjects' behavior in terms of multiple parameters governing the dynamics of a drift-diffusion decision process (*Ratcliff and McKoon, 2008*). We use an enhanced model of the drift-diffusion process so that we can obtain trial-by-trial, moment-by-moment estimates of accumulating evidence (*Brunton et al., 2013*). This model converts the incoming stream of each trial's discrete left and right click stimuli into a scalar quantity $a(t)$ that represents the gradually accumulating difference between the two click streams; each right click increases the value of $a(t)$, whereas each left click decreases $a(t)$. Eight parameters, quantifing sensory and accumulator noise, the leakiness or instability of the accumulation process, a sticky accumulation bound, sensory depression or facilitation, side bias (þ), and lapse rate, govern the dynamics of how $a(t)$ evolves in response to the sensory evidence pulses, and how they are then turned into a binary decision. At the end of each trial's stimulus, the accumulator $a(t_{end})$, together with the parameter þ, drives choices: if $a(t_{end}) > $ þ, the model prescribes 'choose right', whereas if $a(t_{end}) < $ þ, the model prescribes 'choose left'. All of the parameters are estimated by fitting the model to the rat's behavior (Materials and methods).

The original model of *Brunton et al. (2013)* was not constructed to explain different types of side biases, so it had only a single parameter (þ) that could account for such lateralized effects. By adding three more parameters that could cause different types of side biases, fitting the extended model to behavioral data following unilateral inactivations, and asking which parameters are most affected relative to control trials, we can better estimate which particular aspect of the behavior was impacted by unilateral inactivations. The three side bias parameters that we consider, in addition to þ, are: asymmetric sensory input gain, asymmetric sensory input noise, and asymmetric lapse rates (Materials and methods). Considering all four of these side bias parameters in the case of unilateral inactivations of the FOF, we previously concluded that FOF inactivations were consistent with perturbing a process that was not part of evidence accumulation directly, but was instead downstream of the accumulation process and therefore followed it (*Erlich et al., 2015*; *Piet et al., 2017*).

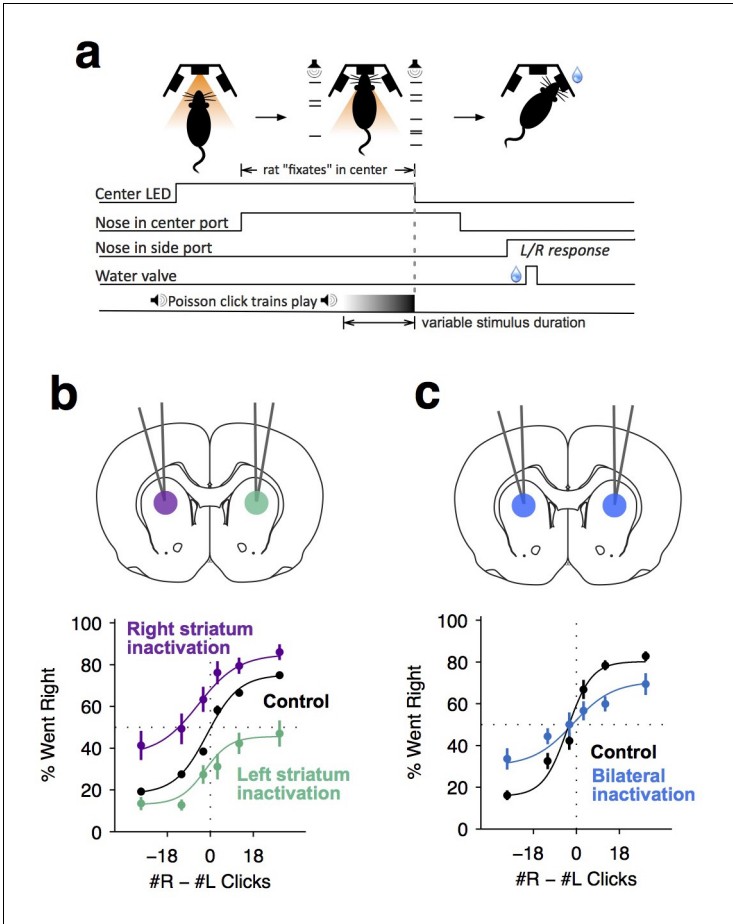

**Figure 1.** Dorsal anterior striatum is required for unimpaired performance on the Poisson-clicks evidence accumulation task. (a) Sequence of events in each trial of the rat auditory Poisson-clicks task. From left to right: after light onset above the center port, rats 'fixate' their position by placing their nose inside the center port. During nose-fixation, two different trains of randomly timed auditory clicks are played concurrently from the left and right speakers. Upon termination of the sound trains, the light above the center port turns off and the rat needs to make a choice, poking into the left or right port to indicate if more clicks were played on the left or right sides, respectively. (b) Unilateral infusion of muscimol into the striatum results in a significant ipsilateral bias on accumulation trials. Purple and cyan psychometric curves show data on days of right and left striatal infusions (n left sessions = 29; n right sessions = 29), respectively. Black psychometric curve shows data from control sessions that occurred one day before infusion sessions (n = 58). (c) Bilateral infusion of muscimol into the striatum results in significant impairment on accumulation trials. The blue psychometric curve is from bilateral infusion sessions (n = 26) and the black psychometric curve is from control sessions that occurred one day before bilateral infusion sessions (n = 26). Data are shown as mean ± S.E.M.

DOI: https://doi.org/10.7554/eLife.34929.002

The following figure supplements are available for figure 1:

**Figure supplement 1.** Rat behavior indicates the accumulation of auditory evidence over the entire trial.
DOI: https://doi.org/10.7554/eLife.34929.003

**Figure supplement 2.** Control LED trials indicate that the behavioral bias and impairments resulting from striatal inactivation are not due to motor impairments.
DOI: https://doi.org/10.7554/eLife.34929.004

Here, we improve upon this analysis and apply it to our striatum inactivation data. At the time of the *Erlich et al. (2015)* study, the complexity of determining the derivative of the model with respect to all 11 of its parameters precluded us from fitting all 11 parameters simultaneously. We instead performed exhaustive scans in the space of two parameters at a time while the other nine parameters were fixed to their control (no inactivation) values (e.g., Figure 4 in *Erlich et al. (2015)*).

Since that time, however, algorithmic differentiation packages, which greatly facilitate computing the derivative of arbitrary differentiable models embodied in computer code, have become widely available (*Abadi et al., 2016*; *Baydin et al., 2015*; *Revels et al., 2016*; *Al-Rfou et al., 2016*). Using the ForwardDiff package of the language Julia (*Revels et al., 2016*) to obtain automatically the derivative with respect to all 11 parameters in the model of *Erlich et al. (2015)*, we constructed a package that can efficiently and simultaneously fit all 11 parameters in the model. We are publishing this package in open source form, as part of the contribution of the current manuscript (code available at https://github.com/misun6312/PBupsModel.jl [*Yoon and Brody, 2018*]; copy archived at https://github.com/elifesciences/PBupsModel.jl). We validated this approach and our previous FOF analysis by fitting all 11 parameters simultaneously to our previous FOF unilateral inactivation data. This new analysis (*Figure 2—figure supplement 2*, *Supplementary file 4*) confirmed the conclusions about the FOF found by *Erlich et al. (2015)*. Following this conclusion, we next turned to performing the same analysis on the inactivation data collected in the current study for the anterior striatum.

For simplicity of presentation, below, we illustrate some of the results of the model fits in terms of psychometric plots (i.e., graphing the probability of a decision to one side as a function of total #R – #L clicks, averaged over trials), but we note again that our model and its fits are sensitive to the detailed timing of the click stimuli in each individual trial, which is information that is obscured in the trial-averaged psychometric plot. As a result, the model and its fits can resolve the effects of different parameters that are indistinguishable in a psychometric plot (see also illustrations of this point in Supplementary Figure S4 in *Brunton et al. (2013)*). For example, a leaky (i.e., forgetful) accumulator and an increased overall lapse rate both predict an overall performance impairment. But the leaky accumulator impairment will be greater for trials that by chance had their clicks earlier rather than later, whereas the lapse impairment will be independent of the timing of each trial's clicks. A model that is sensitive to the timing of each trial's clicks can thus distinguish the two. Similarly, an asymmetric sensory input gain and an asymmetric lapse rate both predict a side bias. But the magnitude of the bias due to an asymmetric input gain will scale with the number of clicks presented on each trial. This contrasts with the bias that would be induced by an asymmetric lapse rate, which would be independent of the number of clicks presented. This again allows the effects of the two parameters to be distinguished. In sum, trial-by-trial and detailed click-timing effects, although not visible in the trial-averaged psychometric plot, impact the likelihood of the data under the model, and thus impact the model fits and the likelihood landscapes (such as those shown in *Figure 2b and d* below). When two parameters trade off in a manner that impairs our ability to distinguish them, this is revealed in the likelihood landscape as a ridge of high likelihood. The shape of the ridge quantifies the extent and scaling of the parameter trade-off (Materials and methods and for example *Figure 2D* in *Brunton et al. (2013)*).

Simultaneously fitting all parameters of the enhanced 11-parameter model to data from sessions with unilateral muscimol inactivation of the anterior dorsal striatum revealed that two parameters differed enough from their control values to produce substantial changes in behavior (*Supplementary file 2*). First, the side bias in the lapse rates (the contralateral lapse rate parameter $\kappa_C$ and the ipsilateral lapse rate parameter $\kappa_I$, which are unitless parameters in terms of fraction of trials; Materials and methods) significantly increased in favor of ipsilateral choices ($\kappa_I$: from 0.29, 95% C.I. = [0.18 0.43] in control sessions to 0.00, 95% C.I. = [0.00 0.14] for inactivation sessions, $\kappa_C$: from 0.20, 95% C.I. = [0.04 0.58] in control sessions to 0.60, 95% C.I. = [0.14 0.93] for inactivation sessions). An effect on lapse rates was also seen after unilateral FOF inactivations, where it was interpreted as an effect on processes subsequent to the accumulator, and not part of it (*Erlich et al., 2015*). Second, the magnitude of the accumulator and sensory noise parameters, which respectively describe diffusion noise intrinsic to the accumulator and noise associated with the addition of each sensory click, also increased significantly (*Figure 2a,b* and *Figure 2—figure supplement 1*). The trade-off between these parameters (*Brunton et al., 2013*) was large enough that it was impossible to distinguish which of the accumulator noise $\sigma^2_a$ or the ipsilateral and contraletral sensory noise parameters $\sigma^2_{s,I}$ and $\sigma^2_{s,C}$ was responsible for the increase. We note that we do not mean to imply that the combination of sensory and accumulator noise is a single, biologically interpretable quantity, but simply that our data cannot distinguish between the different trade-offs between these parameters that fit the data equally well. There was a suggestion that the intrinsic accumulator noise $\sigma^2_a$ specifically increased, with this parameter being significantly greater than zero during inactivation trials whereas it was not distinguishable from zero in control trials (*Figure 2b* and *Figure 2—*

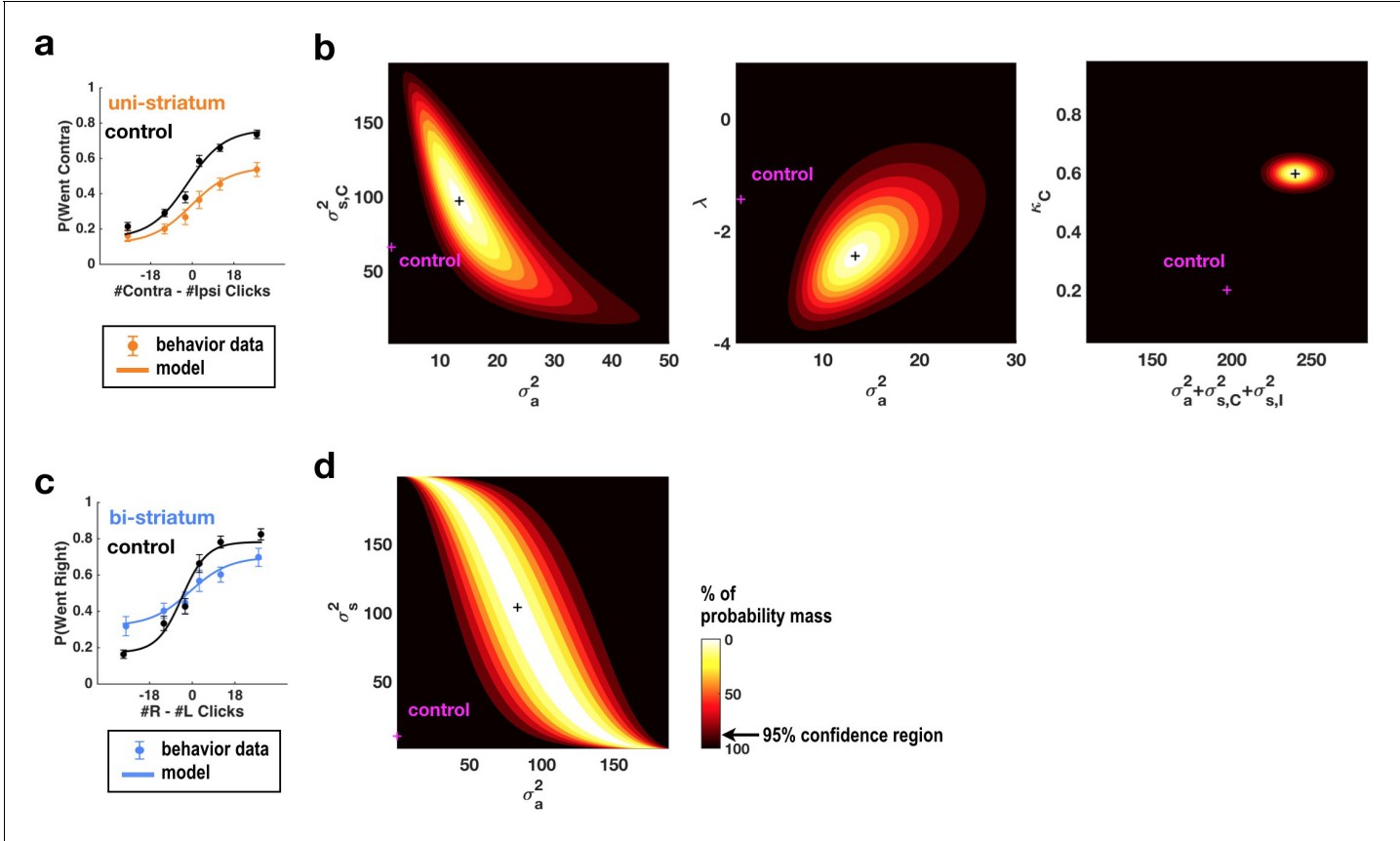

**Figure 2.** Fits of the model of *Brunton et al. (2013)* and *Erlich et al. (2015)* to data from sessions following muscimol inactivation of the striatum. (a) Psychometric curves for control and unilateral inactivation data. Left and right inactivations were collapsed together. Orange data points are from sessions following unilateral infusions of muscimol. The black and orange lines are the psychometric curves predicted from the model fit to the control and inactivation data, respectively. (b) Normalized likelihood of the data given the model, shown as a function of the parameters for which best-fit values for inactivation data were significantly different from best-fit values for control data. Magenta shows the best-fit values for control datas. The black cross shows the best-fit values for inactivation data. The color scale indicates the percentage of probability mass; the region of probability mass >95 indicates the 95% confidence region. Left: sensory noise for the side contralateral to the infusion versus accumulator noise. Although there is a trade-off between accumulator and sensory noise, the weighted sum of the accumulator and biased sensory noise has a best-fit value following unilateral inactivations that is significantly greater than its control best-fit value. Middle: leak/instability parameter versus accumulator noise. Right: summed sensory plus accumulator noise versus lapse rate, which shows that in fact the lapse rate $\kappa_C$ does not trade off with the summed noise. (c,d) As in panels (a,b) but for bilateral striatum inactivation data, and for a model where the sensory noise is constrained to be the same for both sides of the brain, so there is only one sensory noise parameter. Here the tradeoff between sensory noise and accumulator noise is large enough that we cannot distinguish whether one or both are significantly different from their control values, but there is nevertheless a significant increase in their sum.

DOI: https://doi.org/10.7554/eLife.34929.005

The following figure supplements are available for figure 2:

**Figure supplement 1.** Confidence regions with unbounded parameter optimization.
DOI: https://doi.org/10.7554/eLife.34929.006

**Figure supplement 2.** Fitting all parameters simultaneously for unilateral FOF inactivation data confirms the conclusions found by fitting only two parameters at a time.
DOI: https://doi.org/10.7554/eLife.34929.007

**Figure supplement 3.** Psychometric curves of data and simulation data for which sensory and accumulator noise parameters are set to zero.
DOI: https://doi.org/10.7554/eLife.34929.008

**Figure supplement 4.** Psychometric curves of data and simulation data for which the bias parameters are adjusted to control best-fit parameter values in the 11-parameter model.
DOI: https://doi.org/10.7554/eLife.34929.009

*figure supplement 1*), but the difference between control and inactivation $\sigma^2_a$ values was not significant (p<0.15). Lapse rate and noise parameters described distinct effects, and did not trade off with each other (*Figure 2—figure supplement 1a*, third row, middle column).

For data from bilateral inactivation sessions, the combination of sensory and accumulator noise parameters was again significantly greater than for control sessions (*Figure 2c,d*; $w_1\sigma^2_a + w_2\sigma^2_s$: from 31.64 clicks$^2$/sec [95% C.I. = [17.40 55.84]] in control sessions to 117.00 clicks$^2$/sec [95% C.I. = [71.12 156.53]] , where $w_1$ = 0.92, $w_2$ = 0.39 during inactivations. We note that noise magnitudes cannot be less than zero, implying that confidence intervals for both $\sigma^2_a$ and $\sigma^2_s$ are bounded by zero).

These fits contrast with those following FOF inactivation (*Erlich et al., 2015*). In particular, we note that the sensory and accumulator noise parameters were minimally altered after FOF inactivation, whereas ADS inactivation significantly impacted them.

This pharmacological demonstration that the striatum is required for unimpaired decisions that are based on the accumulation of evidence, and the model-based suggestion that the striatum affects properties of the accumulator, led us to explore the detailed neural dynamics that may support its potential causal contribution. To do so, we conducted single-unit recordings from freely behaving subjects engaged in the evidence accumulation task. Consistent with previous work (*Graybiel, 2008*; *Jin and Costa, 2010*; *Kravitz and Kreitzer, 2012*), we found that the neural activity of many striatal neurons was modulated by movement initiation (*Figure 3—figure supplement 1a–c*). However, we also found that over a third of the recorded neurons significantly modulate their activity in a side-selective manner (p<0.05) during the fixation period many hundreds of milliseconds before the movement initiation reporting the decision (64/173 [37%] of the neurons active during the fixation period [*Figure 3—figure supplement 1d–f*]). This timing suggests that they may have a role in forming the upcoming decision (go-left or go-right). These neurons were termed as 'side-selective' and for each we further defined the neuron's prefered side as that yielding the largest activation, as done previously by *Hanks et al. (2015)*. Consistent with previous work in primate dorsal striatum (*Ding and Gold, 2010*), we found that the average responses of these rat striatum neurons ramped upwards for stimuli in the preferred direction (*Figure 3*), and moreover, that after an initial onset latency, the slope of the ramp was proportional to the stimulus strength (*Figure 3*; *Figure 4a*). Importantly, however, a gradual ramping profile is not conclusive evidence for the encoding of gradually accumulating evidence, because such a response profile can also be consistent with other

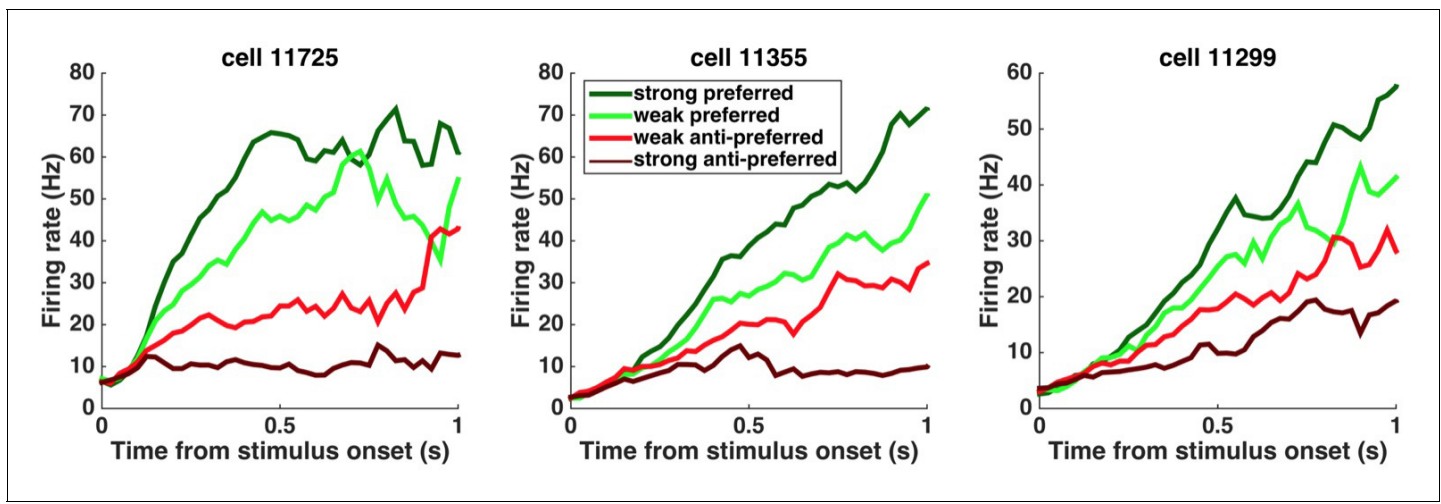

**Figure 3.** Peri-stimulus time histograms (PSTHs) of example neurons. PSTHs aligned to stimulus onset are shown for three example striatum neurons. Trials were sorted into four stimulus-strength bins for each neuron. Green traces correspond to the preferred-direction stimuli and red traces to anti-preferred-direction stimuli. Darker colors correspond to stronger stimuli (less difficult) and brighter colors correspond to weaker stimuli (more difficult).

DOI: https://doi.org/10.7554/eLife.34929.010

The following figure supplement is available for figure 3:

**Figure supplement 1.** Firing rate modulation of striatal neurons.

DOI: https://doi.org/10.7554/eLife.34929.011

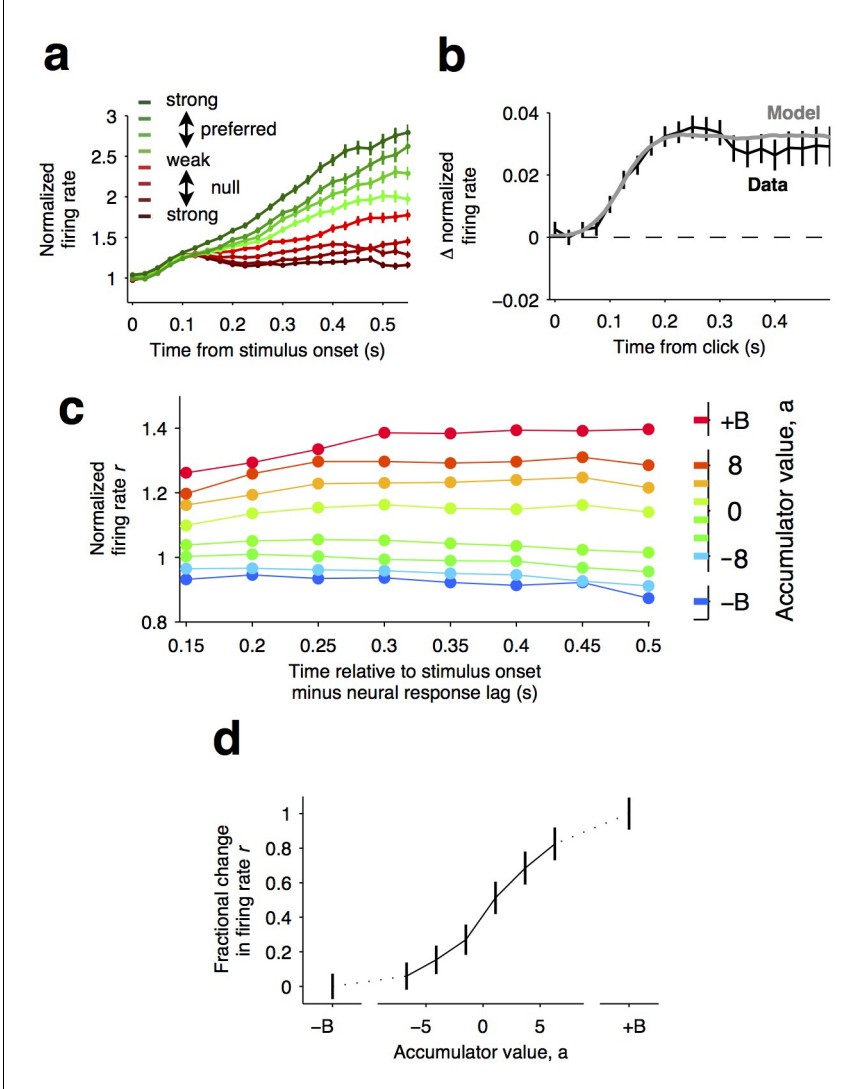

**Figure 4.** Graded representation of accumulated evidence in the dorsal striatum. (**a**) Responses of pre-movement side-selective striatal neurons during evidence accumulation (mean ± S.E.M.). Trials are grouped by the average strength of sensory evidence with greener and redder colors corresponding to stimuli in the preferred and non-preferred direction of the neurons, respectively. Each group of trials is sorted on the basis of the difficulty of the trials from easy to hard, corresponding to darker and lighter colors, respectively. Note the significant dependence of ramping responses on stimulus strength (n = 64 neurons from three rats). (**b**) Click-triggered average response ± S.E.M. Note the close correspondence of the average click-triggered population response to a theoretical prediction of a fixed-magnitude and sustained increase in the neurons' firing rate (see Materials and methods). (**c**) Firing of striatal neurons aligned to trial onset minus the neural response lag (150 ms; see Materials and methods) grouped on the basis of model-derived accumulator value (colors with ± B correspond to sticky accumulation bounds). Note that this accumulator value to firing rate map is graded and fairly stable over time (n = 64 neurons). (**d**) The population change in firing rate as a function of accumulator value averaged across time exhibits a graded response.

DOI: https://doi.org/10.7554/eLife.34929.012

The following figure supplement is available for figure 4:

**Figure supplement 1.** Computing tuning curves that describe the relationship between neural activity and accumulated evidence.

DOI: https://doi.org/10.7554/eLife.34929.013

encoding schemes (*Ditterich, 2006*; *Hanks et al., 2015*; *Latimer et al., 2015*) such as step changes in firing rate that occur at different times in different trials (*Latimer et al., 2015*). Thus, we extended our analysis to include a more direct test in which the influence of single quanta of sensory evidence on the responses of the cells is quantitatively assessed.

If indeed temporal integration underlies the ramping activity of the striatal cells, then each single quantum of sensory evidence (an auditory click) should result in a fixed-magnitude and a sustained increase in the neuron's firing rate (*Figure 4b*, model) (*Hanks et al., 2015*; *Huk and Shadlen, 2005*). We thus estimated the effect of each sensory evidence quantum by computing the click-triggered average response of the side-selective striatal neurons. We found that striatal neurons modulated their activity in close agreement with this theoretical prediction (*Figure 4b*, data), arguing in favor of a role of this anterior striatal subregion in the behavioral accumulation of evidence process.

We also took advantage of a recently developed method, i.e., direct estimates of firing rates as a function of accumulated evidence, to compute neural tuning curves (*Hanks et al., 2015*). Model-derived estimates of the moment-by-moment value of the accumulating evidence in each trial are collated with simultaneously recorded firing rates to generate tuning curves for accumulated evidence (see *Hanks et al. (2015)*), Materials and methods, and the illustration of the method in *Figure 4—figure supplement 1*). When applying this analysis to the striatal data, we found that the side-selective neurons encoded accumulating evidence in a remarkably graded manner throughout the period of evidence accumulation (*Figure 4c,d*). This graded encoding was consistent across different neurons in the population of recorded striatal cells (*Figure 5*). Such a graded representation implies that the striatum carries information about the graded value of accumulated evidence, as would be necessary for a brain structure involved in such a process.

Our pharmacological methods address the questions of *whether* the anterior dorsal striatum is involved in the process of accumulation of evidence, and our electrophysiological and computational methods address *how* the anterior dorsal striatum represents the accumulation of sensory evidence. However, neither directly addresses the question of *when* the anterior dorsal striatum is involved. This question is critical and has proven to be pivotal in assessing the involvement of a brain region in the evidence accumulation process. For example, some brain regions can be required for

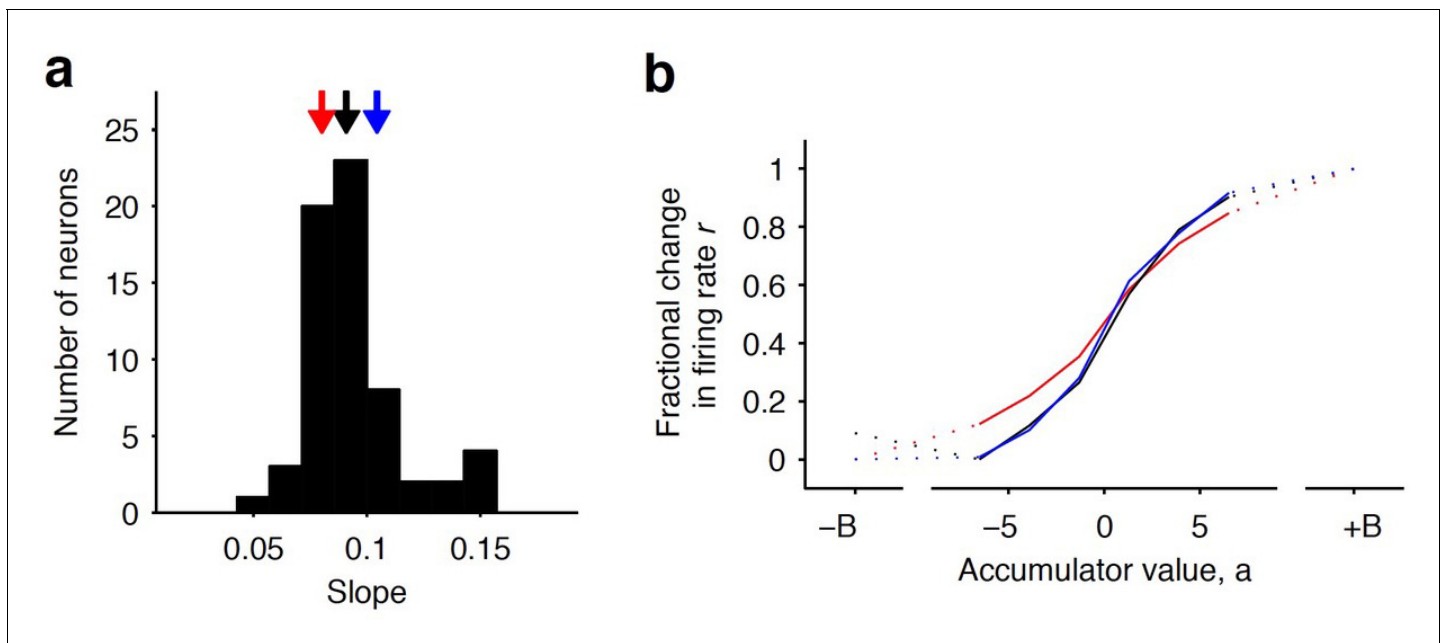

**Figure 5.** Distribution of tuning curve slopes for individual striatal neurons. (a) Histogram of the slope of individual neurons obtained from a sigmoidal fit of the relationship between firing rate and accumulator value. The black arrow indicates the median value of the distribution (50th percentile). Red and blue arrows indicate points corresponding to the 20th and 80th percentile marks, respectively. (b) Example tuning curves shown for 20th, 50th, and 80th (colored as in [a]) percentile neurons. Graded encodings of accumulated evidence are exhibited for all of these neurons.
DOI: https://doi.org/10.7554/eLife.34929.014

decisions that are based on accumulation of evidence, yet contribute at times suggesting that they are instead required for processes that are subsequent to the gradual accumulation of evidence itself (*Erlich et al., 2015*; *Hanks et al., 2015*). No region to date has been reported to be required at points of time that fully coincide with the evidence accumulation period.

To delineate the precise timing of the anterior dorsal striatum's contribution, we used optogenetic inactivation, mediated by halorhodopsin (eNpHR3.0), to unilaterally and transiently inactivate this region during the Poisson Clicks task. We expressed eNpHR3.0 using viral delivery methods (*Figure 6a*; Materials and methods). Acute neural recordings in our experimental rats verified that we could indeed transiently silence neural activity in the striatum with fine temporal precision using the delivery of green light (*Figure 6b*). We began with full-trial unilateral optogenetic inactivation and found, in agreement with the pharmacological inactivation described above, that optogenetic manipulation resulted in more ipsilateral choice biases relative to control trials, which in this case were randomly interleaved with the inactivation trials (*Figure 6c*; bias = 9.0 ± 2.3%, p<0.01). These effects were consistent across rats (*Figure 6d*). Control rats whose striatum was injected with the same virus expressing YFP alone did not show a behavioral bias (bias = 0.1 ± 1.8%, p=0.89). Next, to resolve directly *when* the striatum contributes to the auditory accumulation of evidence task, we transiently inactivated it unilaterally during one of four different 500 ms time periods during the task: (i) the delay period immediately preceding stimulus onset ('pre-accumulation'), (ii) the first half of a 1 s sensory stimulus ('first half'), (iii) the second half of a 1 s sensory stimulus ('second half'), or (iv) the movement period ('post-choice'). In contrast to similar inactivation assays of the cortical FOF, which have no effect during the early parts of the accumulation period (*Hanks et al., 2015*), we found that transient optogenetic inactivation of the anterior dorsal striatum during both the first half and second half of the accumulation caused a significant bias for the ipsilateral choices, with a similar magnitude of effect in these two periods (first half bias = 10.4 ± 4.0, p<0.01; second half bias = 12.9 ± 3.7%, p<0.01; difference = 2.5 ± 2.8, p=0.2; *Figure 6e*; the first-half effect in the striatum is significantly greater than that in the FOF, p<0.01, *Figure 7*). Remarkably, the effect in striatum was limited to the stimulus presentation period and we found no significant effect of optogenetic inactivation during the pre-accumulation or post-choice periods (pre-accumulation bias = 0.4 ± 5.4%, p=0.42; post-choice bias = 0.9 ± 5.2%, p=0.38; *Figure 6e*). These results are consistent with the idea that the anterior dorsal striatum plays a direct causal role throughout the entire evidence accumulation process.

## Discussion

Studies carried out over more than two decades have attempted to elucidate neural circuits that underlie the accumulation of evidence over time (starting with *Shadlen and Newsome (1996)*; see *Gold and Shadlen (2007)*, *Carandini and Churchland (2013)*, *Brody and Hanks (2016)*, and *Hanks and Summerfield (2017)* for reviews; see also *Carandini and Churchland (2013)*, *Gold and Shadlen (2007)*, *Krajbich et al. (2012)*, *Shadlen and Newsome (1996)*). Ding and colleagues have shown that microstimulation of the ADS perturbs decisions based on the accumulation of evidence (*Ding and Gold, 2010, 2012a*; *Ding, 2015*; but see *Histed et al., 2009* and *Tehovnik and Slocum, 2013*) for discussion as to whether or not microstimulation primarily affects axon terminals, which would add complications for its interpretation when localizing neural function). Despite these many years of studies, no brain region has previously been identified as: first, being required for unimpaired accumulation-based decision-making behavior; second, having the graded neural encoding required for direct involvement in computing the graded, gradually evolving, value of the accumulating evidence; and third, making a causal contribution throughout times that fully coincide with the accumulation process. By demonstrating that the anterior dorsal striatum satisfies all three of these criteria, our work suggests that the anterior dorsal striatum is the first identifiable node in the neural circuit causally responsible for computing evidence accumulation. The anterior dorsal striatum is well positioned anatomically to participate in evidence accumulation as it receives diverse convergent anatomical input from multiple cortical areas (*Cheatwood et al., 2003*; *McGeorge and Faull, 1989*) and it is connected via recurrent loops with cortical and subcortical areas that are widely believed to play a role in action selection (*Ding and Gold, 2013*). Whether the anterior dorsal striatum possesses a unique role in evidence accumulation, or whether it is an important node of a more extended network of brain regions that operate in coordination to

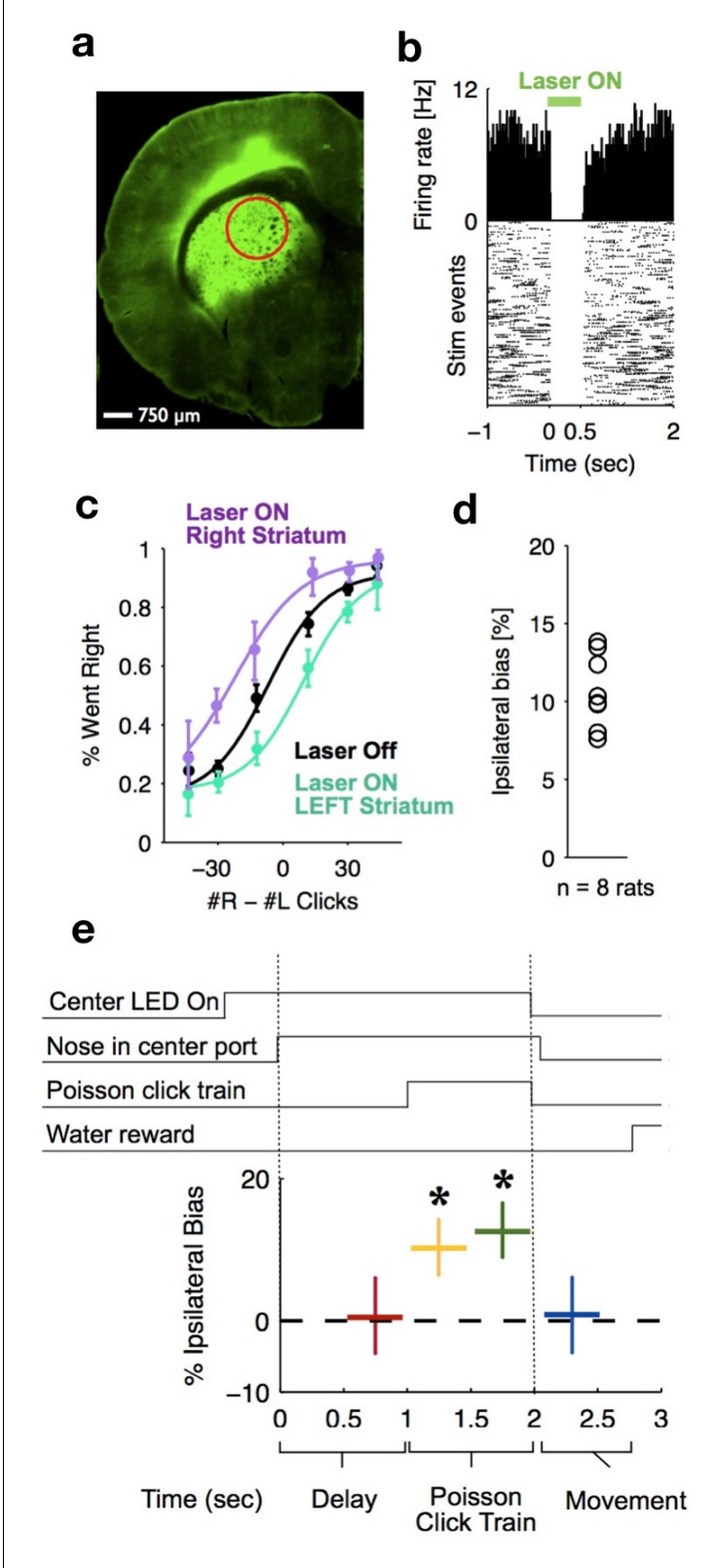

**Figure 6.** Optogenetic inactivation reveals that dorsal striatal activity causally contributes to decision formation throughout the accumulation process but not before nor after. (a) Coronal section of the left hemisphere showing the expression of eYFP-eNpHR3.0 in the left dorsal striatum. Optical fiber localization and 750 μm estimated inactivation radius are indicated by the red circle. (b) Raster plot (bottom) and peri-stimulus time histogram (top)
*Figure 6 continued on next page*

*Figure 6 continued*

showing the effectiveness in silencing of local striatal activity in response to delivery of green light (indicated by the green bar at the top). (c) Unilateral full-trial optical inactivation of the striatum results in an ipsilateral bias in accumulation trials. The purple and cyan psychometric curves show data for right and left striatal inactivation, respectively, whereas the black psychometric curve shows data from control trials that occurred on the same days (n = 8 rats). (d) Scatter plot indicated the mean ipsilateral bias for each individual rat. (e) Bottom: behavioral bias caused by 500 ms inactivation during the pre-stimulus delay period (red), the first half of the sensory stimulus (yellow), the second half of the stimulus (green) and upon initiation of movement (blue). Top: task structure. Note the significant effect (indicated by an asterisk) only during evidence accumulation but not prior to the presentation of sensory stimuli nor after.

DOI: https://doi.org/10.7554/eLife.34929.015

mediate evidence accumulation, remains to be resolved. Corticostriatal loops are organized as distinct parallel circuits (*Alexander et al., 1986*; *Kim and Hikosaka, 2015*); future studies dissecting the contribution of different loops will be important for resolving this major question.

Our results, together with those of Ding and colleagues (*Ding and Gold, 2010*; *Ding and Gold, 2012a*; *Ding, 2015*), suggest that the striatum may be directly involved in a more expansive set of computations, traditionally considered to be more cognitive in nature, than the already well-established functions of the dorsal striatum in action selection, response initiation, evaluation of reward uncertainty, and habit formation (*Ding and Gold, 2010*; *Graybiel, 2008*; *Hikosaka et al., 2014*; *Jin and Costa, 2010*; *Kravitz and Kreitzer, 2012*). It will be important to better understand how the striatal involvement in computing accumulation of evidence, as identified in this study, may contribute to those previously established functions. Extensions to our paradigm (for example, free response protocols or more extended trial durations) are likely to be useful for reconciling the functions indciated by results with the other functions of the striatum. The computations involved in evidence accumulation may perhaps provide an efficient mechanism for extracting important pieces of information from the environment in the service of other roles of the striatum.

By identifying model parameters affected by the inactivations, our model fits suggest specific aspects of the evidence accumulation computation that could be prioritized as potentially particularly strongly related to the ADS's role in the computation. The model fits to unilateral pharmacological inactivation data found that, similar to unilateral inactivations of the FOF, the side bias in lapse rates was increased by the inactivations. But in contrast to what occurs in the FOF, noise parameters, including the accumulator's intrinsic noise, were also substantially increased after striatal inactivation (*Figure 2a,b*). Model fits to bilateral anterior dorsal striatum inactivation data found that the sum of sensory and accumulator noise magnitude parameters was significantly increased by striatal inactivation (*Figure 2c,d*). A parsimonious account suggests that the main noise parameter that is affected may perhaps be the magnitude of the noise in the evidence accumulator. This would be consistent with the idea, supported by our electrophysiological and optogenetic data, that the striatum plays a role in the

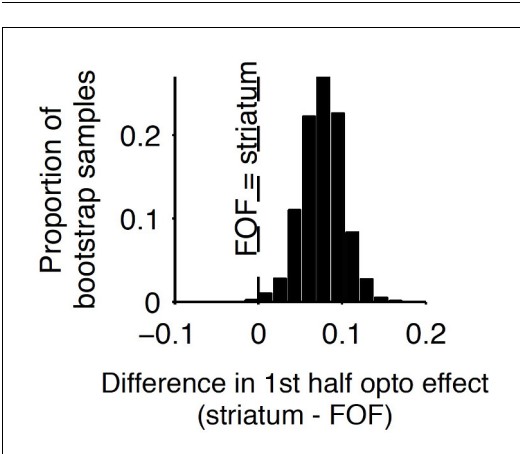

**Figure 7.** Comparison of early stimulus period optogenetic inactivation effects in the striatum and frontal orienting field (FOF). Optogenetic inactivation of the anterior dorsal striatum during the first half of the 1 s stimulus presentation period produced a significantly larger effect than the same manipulation of the FOF (p<0.01), with the latter data coming from a previous report. For this analysis, individual trials were resampled with replacement from both data sets across 1000 iterations, and the difference in inactivation effect was calculated for each iteration to provide a nonparametric statistical comparison. As reported above, the first-half anterior dorsal striatum effect itself is significant, and as reported previously, the first-half FOF effect is not significant, but a direct comparison as described here is still necessary to establish a significant difference.

DOI: https://doi.org/10.7554/eLife.34929.016

accumulation process. The lack of a significant effect on other parameters should be treated with caution: it remains possible that future studies with greater statistical power could discern an effect of striatal inactivation on some of these other parameters. Nevertheless, even while we emphasize that we do not take the modeling results on their own as conclusive, they do suggest that accumulator noise is a principal parameter of interest. It is conceivable that bilateral striatal inactivation increases accumulator noise by destabilizing the accumulator's representation without biasing it, but a circuit model hypothesis to explain precisely how the striatum might affect the accumulator noise level remains to be developed. Another important direction for future studies will be the development of models with temporally specific parameters that could be used to model the effects of temporally specific optogenetic inactivation appropriately.

Independently of whether the anterior dorsal striatum operates alone or as part of a broader circuit for computing gradual evidence accumulation, and independently of the precise nature of its contribution to the evidence accumulation computation, the data reported here provide a critical foothold towards delineating the relevant causal circuit: for example, the anterior dorsal striatum's major inputs and outputs become important candidate regions to be examined for a potential role in the process. The possibility that the causal circuit for computing evidence accumulation may be delineated in the near future suggests that we will soon be able to elucidate the circuit and cellular mechanisms that support evidence accumulation, a computation that is crucial for decision-making behavior in a wide range of species, including humans.

## Materials and methods

### Subjects and housing

All animal procedures described in this study were approved by the Princeton University Institutional Animal Care and Use Committee and were carried out in accordance with National Institutes of Health standards. All subjects were adult male Long-Evans rats (Taconic, NY) that were pair housed in Technoplast cages and were kept in a 12 hr reversed light-dark cycle. All training and testing procedures were conducted during the dark cycle. Rats had free access to food but had restricted water access. The amount of water that the rats could obtain daily was limited to 1 hr per day of free water (starting 30 min following the end of training), in addition to what they could earn during training.

### Behavior

Rats were trained seven days a week at similar times each day for a period of about 110 min daily. The training took place in custom-made training boxes (Island Motion, NY) placed inside sound- and light-attenuated chambers (H10-25A, Coulbourn Instruments, PA). Each box consisted of three straight walls and one curved wall in which three nose ports were embedded (one in the center and one on each side, *Figure 1a*). Each nose port also contained one light-emitting diode (LED) that was used to deliver visual stimuli, and the front of the nose port was equipped with an infrared (IR) beam to detect the entrance of the rat's nose into the port. A speaker was mounted above each of the side ports and was used to present auditory stimuli. Each of the side ports also contained a sipper tube that was used for water reward delivery, with the amount of water controlled by valve opening time.

All rats were trained using a semi-automated training protocol on a previously developed accumulation of evidence task (*Brunton et al., 2013*). Training and testing procedures were similar to those described previously (*Erlich et al., 2015*; *Hanks et al., 2015*). In brief, at the start of each trial, rats were instructed to place their nose in the central port and to maintain nose fixation in response to LED illumination of that port. Subsequently, after a delay of at least 200 ms, rats were presented with a two trains of auditory clicks presented simultaneously, one from the left and one from the right speaker. For neurophysioloigcal recordings and pharmacological (muscimol) inactivation experiments, the click train duration varied between 0.1 to 1.2 s. For optogenetic experiments, the stimulus duration was fixed at 1 s for all trials. The train of clicks from each speaker was generated by an underlying Poisson process, with different mean rates for each side. The combined mean click rate was fixed at 40 Hz, and trial difficulty was manipulated by varying the click rate ratio between the two sides. The mean click rate ratio varied from 39:1 clicks/s (easiest) to 26:14 (most difficult). Upon completion of stimulus presentation, the central LED was turned off and rats had to orient towards

the side that played more clicks and nose poke into the corresponding port to obtain water reward of 24 µL.

## Behavioral model

To quantify the animals' behavior and the effect of infusions, we used the behavioral model of *Brunton et al. (2013)* and *Erlich et al. (2015)*), who described it in detail. As in previous work (*Erlich et al., 2015*), to obtain a sufficient number of perturbation trials for the model-based analysis, we combined data across rats and assessed the effects and their statistical significance on this 'meta-rat' data. In *Supplementary file 1*, we also present the result of fits to each individual rat, which are in general consistent with the meta-rat conclusions. We present an abbreviated description here. The model converts each trial's incoming stream of discrete left and right clicks into an accumulating evidence quantity *a(t)* that determines choice behavior. Parameters that govern how *a (t)* evolves are fitted on the basis of the rat's behavior, with the choices made in individual trials constraining the possible trajectories of *a(t)*. Thus, in each trial, the model estimates the evolution of a noise-induced probability distribution over values of the accumulating evidence *a(t)*. To evaluate how well a particular set of parameter values $\theta$ fits the behavioral data, we computed the probability of observing the data given the model. Let $t_{i,R}$ and $t_{i,L}$ represent the right and left click times on trial *i*, let *D* represent the subject's decision in trial *i*, and let *D* present the full set of the subject's decisions across all trials. Assuming that trials are independent, we may compute the likelihood of the model:

$$P(D|\theta) = \prod_i P\big(d_i|t_{i,R},\ t_{i,L},\theta\big)$$

The best-fit parameter values (also known as the maximum likelihood values) are the parameters that maximize the likelihood.

### Dynamics of *a(t)* in each trial

We discretize both time *t* and space *a*, and for each trial, start with the distribution at time *t* = 0, which we set to be a delta function at *a* = 0. We then compute the probability distribution for the next time step given the probability distribution for the previous timestep, and iterate through timesteps until t = T, where T is stimulus duration. At each time point, the accumulator memory *a(t)* represents an estimate of the right versus left evidence accrued so far. At stimulus end, the model decides right if $a > þ$ and left otherwise, where þ is a free parameter. Right (left) pulses change the value of *a* by positive (negative) impulses of magnitude *C*. $\sigma_a^2$ is a diffusion constant, parameterizing noise in *a*. $\sigma_s^2$ parameterizes noise when adding the evidence from a right or left pulse: For each click, variance $\sigma_s^2$ is scaled by the amplitude of *C* and then added to the evidence contributed by the click. λ parameterizes consistent drift in the memory *a*. In the 'leaky' or forgetful case (λ <0), drift is toward *a* = 0, and later pulses affect the decision more than earlier pulses. In the 'unstable' or impulsive case (λ >0), drift is away from *a* = 0, and earlier pulses affect the decision more than later pulses. The memory's time constant τ = 1/λ. B is the height of the sticky decision bounds and parameterizes the amount of evidence necessary to commit to a decision. $\phi$ and $\tau_\phi$ parameterize sensory adaptation by defining the dynamics of *C*. Immediately after a click, the magnitude *C* is multiplied by $\phi$. *C* then recovers toward an unadapted value of 1 with time constant $\tau_\phi$. Facilitation is thus represented by $\phi$ >1, whereas depression is represented by $\phi$ <1.

These properties are implemented by the following equations:

if $|a| \geq$ B then $\frac{da}{dt} = 0$; else

$$da = \sigma_a dW + (\delta_{t,t_R} \cdot \eta R \cdot C - \delta.t, t_L \cdot \eta L \cdot C)dt + \lambda a dt$$

where $\delta_{t,t_{R,L}}$ are delta functions at the times of the pulses; $\eta$ are Gaussian variables drawn from $N(1, \sigma_s)$; and *dW* is a white-noise Wiener process. The initial condition *a(t* = 0) is 0.

Adaptation dynamics are given by:

$$\frac{dC}{dt} = \frac{1-C}{\tau_\phi} + (\phi - 1)C\big(\delta_{t,t_R} + \delta_{t,t_L}\big)$$

In addition, a lapse rate κ parameterizes the fraction of trials in which a random response is made. Ideal performance (a = #right clicks − #left clicks) would be achieved by:

$\sigma_a^2 = 0, \; \sigma_s^2 = 0, \; \phi = 0, \; \text{þ} = 0, \; \kappa = 0$

A list of this model's parameters is thus:

λ – quantifies leakiness (drift toward a = 0) or instability (drift away from a = 0) in the memory of the accumulated evidence. λ is equivalent to 1/τ, where τ is the time constant of the accumulator's memory.

$\sigma_a^2$– a diffusion constant ('accumulator noise'), quantifies random noise in the accumulator's memory that is independent of the sensory stimuli.

$\sigma_s^2$ – 'sensory noise', quantifies noise introduced into the accumulator with each sensory stimulus pulse.

B – a 'sticky commitment bound' (*Ratcliff and McKoon, 2008*), interpreted as the magnitude |a| at which the subject would make and commit to a decision.

φ – the sensory adaptation parameter, quantifying whether successive clicks depress (φ <1) or facilitate (φ >1).

$\tau_\phi$– the time constant of recovery from adaptation.

þ - 'accumulator threshold bias', quantifies whether the final value of *a(t)* has a tendency to be classified as 'go right' (þ <0) or 'go left' (þ >0).

κ - the 'lapse rate', or fraction of trials in which the subject ignores the stimulus and randomly chooses right or left.

## Complexity of the model

Each term in the model describes a distinct effect (*Brunton et al., 2013*), and is statistically significant, with the exception of one parameter, B, which is not needed here: it is not affected by ADS perturbations, and confidence intervals for B include B = infinity for all rats in all experiments (see *Supplementary file 1*). B = infinity corresponds to not having that parameter at all.

In addition, we have used the Bayesian Information Criterion (BIC) (*Nagin, 1999*), which takes into account and penalizes model complexity, to assess whether any parameter could be eliminated from the model (see *Supplementary files 6* and *7*). The only parameter that can be eliminated across all conditions is once again the parameter B.

Thus, except for the accumulator bounds parameter B, which we keep for consistency with other papers, both BIC and confidence intervals indicate that we should keep all parameters. We have nevertheless chosen to also keep B, rather than to eliminate it from the study, so as to document that although striatum perturbations could, in principle, have affected B, in fact they did not.

## Model-fitting

We used the procedures detailed in *Brunton et al. (2013)* and *Erlich et al. (2015)*, except that in this work we used an algorithmic differentiation package, the ForwardDiff package in the language Julia (*Revels et al., 2016*), to obtain the derivative with respect to all model parameters automatically. We have published the model-fitting code in open source form (https://github.com/misun6312/PBupsModel.jl).

In brief, the model equations described above also determine how a probability distribution of values of a evolves over time. We do not use Monte Carlo simulations to approximate the probability distribution, but instead, for each trial given its individual click times, we simulate the time-evolution of this probability distribution directly. The probability distribution of a at the end of the trial $t_{end}$ then defines the probability of going Right: $P(a, t_{end}) \geq$ þ , and the probability of going Left: $P(a, t_{end}) <$ þ . Thus, given the model parameters, we can obtain the probability in each trial of observing the choice that the subject made. As described above, assuming that different trials are independent of each other, the probability of seeing the subject's full data set is the product of all the trial probabilities. This net probability is what we refer to as the likelihood, and we optimize parameter values to find the highest likelihood. Under a 'flat prior' assumption, that is, when we make no prior assumptions about parameter values, the likelihood can then be normalized to correspond to the probability of seeing certain parameter values given the data, and is used to determine confidence regions for the parameters.

Each model fit was run 10 times, starting from different parameter value seeds. Consistent with previous similar fits (see Supplementary Fig. S6 in *Brunton et al. (2013)*), all fits ended within the

final confidence region. Of the ten fits, the one with the highest log likelihood (and a positive semi-definite Hessian) was chosen as the one that defined the best-fit parameter values and the confidence region around them.

## Trade-offs between different model parameters

Trade-offs between parameters are quantified by ridges of high likelihood in the likelihood landscapes (see *Figure 2* in *Brunton et al. (2013)* and *Figure 2* in the main text). The orientation of the ridge quantifies the parameter-to-parameter scaling that best trades parameters off against each other; and the extent of the ridge quantifies the range of the trade-off for which model fits are almost as good as the maximum likelihood fit. Such ridges can be examined by plotting the likelihood of the data as a function of pairs of model parameters (as in *Figure 2*). But a full description of the trade-offs across all $N$ parameters involves describing the shape of the $N$-dimensional maximum likelihood peak. Given sufficient numbers of trials, this peak is well-approximated by an $N$-dimensional Gaussian (*MacKay, 2003*); the covariance matrix $C$ of this Gaussian (corresponding to the inverse of the Hessian of the log likelihood, which we compute not numerically but with algorithmic differentiation, which is much more precise) is then the second-order approximation that describes the shape of the maximum likelihood peak (MacKay 2003). We focus on the eigenvector of $C$ with the largest eigenvalue; this eigenvector describes the orientation of the major ridge. Its corresponding eigenvalue quantifies the variance along this ridge, and thus describes the extent of the trade-off region.

## Unbounded parameter optimization

Some of the model parameters have natural range constraints on their ranges; for example, noise magnitudes are bounded by zero since the variance cannot be less than zero. However, optimization procedures, as well as the Gaussian approximation that describes the shape of the likelihood landscape around its optimum, are simpler in unbounded spaces. Consequently, for each bounded parameter $\alpha$, we defined a bounded monotonic nonlinear function f such that $\theta = f(\alpha')$ remained within the desired bounds even when $\alpha'$ was unbounded. We carried out the optimization using the unbounded $\alpha'$ parameter (which we refer to as 'infinite space' since $\alpha'$ has infinite range), and computed confidence intervals in the infinite space. We then used the f function to compute the corresponding optimal value of $\alpha$ and confidence intervals for $\alpha$. (See, for example, *Figure 2—figure supplement 1*). When the optimal value of a parameter sits at the boundary of the range, the true optimal point can exist outside the boundary. This could have an impact on other parameters that are interacting with the parameter. Also, it may cause the Hessian matrix at the optimal point to be not positive semidefinite. To resolve this issue, we used the tanh function to convert the parameters to the unbounded domain and then we run optimization in unbounded space. After we find the optimal point, the parameters were transformed to the original coordinates. With this unbounded minimization method, we could obtain the positive semidefinite Hessian matrix, which has all nonnegative eigenvalues.

## Four model parameters to quantify sources of a lateralized bias

The original model (*Brunton et al., 2013*) had only a single parameter that could describe a right versus left choice bias, the decision borderline Þ. By adding three more parameters that could cause different types of side biases, fitting the extended model to behavioral data following a unilateral inactivation, and asking which parameters are most affected relative to control trials, we can estimate which particular aspect of the behavior was impacted by the inactivation. *Erlich et al. (2015)* fit each of these parameters individually. Here, using algorithmic differentiation, we fit all 11 parameters simultaneously. In all, the four different sources of a lateralized choice bias that we considered were:

### Accumulator threshold bias (þ)

The accumulator is categorized into 'Go Left' vs 'Go Right' decisions by comparing the accumulator's value to þ. In the behavioral model, this is implemented by setting the decision borderline to Þ. At stimulus end, the model decides right if $a > þ$ and left otherwise, where þ is a free parameter. It

quantifies whether the final value of *a(t)* has a tendency to be classified as 'go right' (þ <0) or 'go left' (þ >0).

## Post-categorization bias ($\kappa_C - \kappa_I$)

When performing unilateral inactivation, the choice directions can be mapped as 'contralateral' or 'ipsilateral' with respect to the side of inactivation. Contralateral lapse rate is a fraction of the trials categorized as choices contralateral to the inactivated side of the brain, and converts them into ipsilateral choices. And ipsilateral lapse rate is the fraction of the trials categorized as choices ipsilateral to the inactivated side of the brain, and converts them into contralateral choices. The scaling is biased when $\kappa_C \neq \kappa_I$. So, we re-parametrize lapse rate parameters for each side as a total lapse ($\kappa_C + \kappa_I$) and a biased lapse ($\kappa_C - \kappa_I$).

## Input gain bias (gw)

This can be thought of as a form of sensory neglect: Left and Right clicks have different impact magnitudes on the value of the accumulator. 0.5 is the balanced point, where left and right clicks have same impact magnitudes. If the value of input gain weight is lower than 0.5, then ipsilateral clicks have a stronger impact, and decision will consequently be biased to the ipsilateral side. The closer the value to 0, the stronger the impact of ipsilateral clicks. Whereas, the closer the value to 1, the stronger the impact of contralateral clicks.

The magnitude of the input *C* is given by:

$C = 2 * gw * C_i - 2 * (1 - gw) * C_c$, where *gw* is the input gain bias, $C_i$ is the sum of ipsilateral clicks and $C_c$ is the sum of contralateral clicks.

## Biased sensory noise (separate $\sigma_{s,L}^2$ and $\sigma_{s,R}^2$)

By differentially affecting signal-to-noise ratios from the two sides, biased sensory noise can be thought of as a form of sensory neglect distinct from input gain bias: Left and Right clicks have different magnitudes of noise in their impact. The biased sensory noise was implemented by allowing the contralateral noise variance to be a free parameter, fit to the behavioral data from unilateral inactivation trials.

# Surgery

The experiments described in this manuscript focus on the anterior dorsal striatum of the rat at stereotaxic coordinates of 1.9 mm anterior and 2.4 mm lateral, relative to bregma. Each rat received one of three surgical procedures that have all been described in detail elsewhere for different brain areas but were identical in all other respects. These were: (i) implantation of a tetrode-based microdrive consisting of eight tetrodes (three rats, left striatum) (*Erlich et al., 2011, 2015*; *Hanks et al., 2015*), (ii) cannulas for pharmacological inactivation (four rats, bilateral) (*Erlich et al., 2011*; *Hanks et al., 2015*) and (iii) chemically etched optical fibers coupled with viral injection (*Hanks et al., 2015*) (13 rats; six left striatum and seven right striatum). The injected virus consisted of 2–3 μL of AAV virus (either AAV5-CaMKIIα-eYFP-eNpHR3.0 or AAV5-hSyn-eYFP-eNpHR3.0 or a mixture of both at a ratio of 1:2, respectively). Two of the three rats that were used for electrophysiological recordings and received a tetrode implant targeting the anterior dorsal striatum were further injected with AAV5-CaMKIIα-eYFP-ChR2 and were implanted with two optical fibers and an additional tetrode-based microdrive targeting the rat SNr, GPe and superior colliculus, respectively. These data are not discussed in the present manuscript. The infusion cannulas were implanted at an angle of 15° lateral to minimize any potential backflow of muscimol to the frontal orienting fields (FOF), which have recently been demonstrated to be necessary for maximal performance on this task (*Erlich et al., 2015*). Accurate placement of all implants and viral injection targeting was verified histologically.

# Infusions

Infusion procedures follow methods described in detail previously (*Erlich et al., 2015*). Briefly, infusions were generally performed during normal training sessions, were usually at least one week apart, and were never on consecutive days. Control sessions took place on the day prior to the infusion session. On the day of infusion, rats were lightly anesthetized with 2% isoflurane and anesthesia was sustained via continuous delivery of isoflurane using a nose cone. Using a Hamilton syringe that

was attached via tubing to the injector, we delivered 0.5 μL of muscimol at a concentration of 0.125 mg/mL to either the left or right side of the anteriodorsal part of the rat striatum during unilateral infusion sessions and to both sides during bilateral infusion sessions (*Figure 1b and c*, respectively). After delivery, the injector was left inside the brain for a minimum period of 5 min to allow adequate diffusion before removal and also to minimize backflow along the cannula tract. Subsequently, the injector was removed, the cannula was closed, and the rat was removed from isoflurane anesthesia and placed back into its home cage. We allowed 30 min of recovery from anesthesia before placing the rat into the behavioral box. The task performed by the rats during infusion and control sessions was identical to that described above with the exception that on ~10% of trials, rats were presented with an LED above the right or left port and had to orient towards that port. Thus, the animals had to perform the same motor action but success was not dependent on evidence accumulation, hence this procedure controlled for any potential motor deficits that could have arisen due to the inactivation procedure that could impair the animal's ability to orient effectively (*Figure 3—figure supplement 1*).

## Optogenetic perturbation

The methods used in this study for optogenetic perturbation are identical to those described in detail previously (*Hanks et al., 2015*). Prior to each experimental session, a 532 nm green laser (OEM Laser Systems) was connected via a 1 m patch cable with a rotary joint (Princetel) and an FC connector to the rat's optical implant. The rotary joint was mounted on the ceiling of the behavioral chamber. The laser operated at 25 mW and was triggered by a 5V transistor-transistor logic (TTL) pulse, delivered in response to behavioral events and triggered by the automated traiwhayning software (BControl). On all experimental days, laser illumination occurred during a random subset (25%) of trials and was applied unilaterally. Laser illumination trials could be divided into two main types. In the first type, we delivered light for a continuous period of 2 s, starting 500 ms prior to the initiation of the auditory clicks stimulus and ending 500 ms after the termination of the click train. This trial type is defined as 'full-trial' inactivation. For this we used a cohort of eight rats. In the second trial type, which we refer to as 'time-resolved' inactivation, light illumination was delivered in one of four different 500 ms time periods: the delay before stimulus onset, the first half of the 1 s auditory stimulus, the second half of the 1 s auditory stimulus, or during the movement period (*Figure 6e*). All time-resolved inactivation periods were randomly interleaved within single behavioral sessions. For 'time-resolved' inactivation experiments, we used a cohort of seven rats, two of which also belonged to the full-trial inactivation cohort.

The physiological effect of eNpHR3.0 on local neuronal activity was tested using acute recordings in experimental rats (*Figure 6a*), as described previously (*Hanks et al., 2015*). Rats were anesthetized using isoflurane and a sharp etched optical fiber was inserted into the center of the field of viral infection. The optical fiber was coupled with a 532 nm green laser with ~25 mW light intensity at the tip. In parallel, a sharp tungsten electrode (1 MΩ) was positioned adjacent to the optical fiber tip. The effect of laser activation on spontaneous activity was tested by delivering a series of pulses, of 500 ms duration each, at 25 mW every 5 s. The signals from the electrode were amplified, filtered (300–6000 Hz), thresholded on the basis of voltage (30 μV) and sampled at 30.3 kHz (0.25 ms before the threshold triggering and 0.75 ms after; Neuralynx Cheetah). The spikes and TTL pulses were time-stamped with the same 1-MHz clock (Digital I/O, Neurlaynx).

## Histology

The rat was fully anesthetized with 0.4 mL ketamine (100 mg/ml) and 0.2 mL xylazine (100 mg/ml) IP, followed by transcardial perfusion of 100 mL saline (0.9% NaCl, 0.3x PBS, pH 7.0, 0.05 mL heparin 10,000 USP units/mL), and finally transcardial perfusion of 250 mL 10% formalin neutral buffered solution (Sigma HT501128). The brain was removed and post fixed in 10% formalin solution for a minimum period of 7 days. 100 micrometer sections were prepared on a Leica VT1200S vibratome, and mounted on Superfrost Pus glass slides (Fisher) with Fluoromount-G (Southern Biotech) mounting solution and glass cover slips. Images were acquired on a Nikon Eclipse Ti fluorescence microscope under 4x magnification.

## Neural recording and spike sorting

Neural recordings and spike sorting methods have been described in detail previously (*Erlich et al., 2011*; *Hanks et al., 2015*). Briefly, over the course of ~2–4 weeks following surgery, the tetrodes were slowly lowered towards the dorsal part of the rat anterior striatum. On most recording days, an electrically quiet electrode was used as a reference channel, and in the cases where such an electrode was not available, we used the ground of our neurophysiology recording system (Nerualynx) for reference. During recording, a unity-gain preamplifier (HS-32, Neuralynx) was attached to a connector on top of the microdrive via a light-weight tether. Signals from each of the channels were amplified (1,400–5,000×) and band-pass filtered (300–6,000 Hz; Digital-Lynx, Neuralynx). A voltage threshold (20–50 µV) was used for collecting 1 ms spike waveforms, which were sampled at 30.3 kHz (0.25 ms before the triggered event and 0.75 ms after; Neuralynx Cheetah). Neural activity was recorded daily during behavioral sessions that lasted 2–4 hr on average. Regardless of the quality of the recordings, tetrodes were never kept in the same position on different days, and were always moved at the end of each recording day (40–200 µm daily), in order to obtain recordings from new ensembles of neurons daily.

## Analysis of causal perturbation data – optogenetics and pharmacological inactivation

Detailed methods for generating psychometric curves and estimating biases resulting from inactivation in rats performing this exact behavioral task were described recently (*Erlich et al., 2015*; *Hanks et al., 2015*).

In brief, for muscimol inactivation experiments, psychometric curves were generated by concatenating data across either infusion or control sessions for individual rats and by fitting a four-parameter sigmoid described using the following formula:

$$y = y_0 + \frac{a}{1 + e^{-(x-x_0)/b}}$$

In that equation, the 'x' variable is the click difference on each trial (# Right Clicks – # Left Clicks), 'y' is the proportion of trials on which the animal went 'Right', and the four parameters of the fitting procedure are: '$x_0$', the inflection point of the sigmoid; 'b', the slope of the sigmoid; '$x_0$' and 'a + $y_0$', the minimum and maximum of the proportion of trials in which the rat went 'Right', respectively.

For optogenetic inactivation experiments, we measured behavioral bias resulting from transient inactivation of neural activity in a subset of trials (25%) by first binning the trials on the basis of stimulus strength. We then computed the mean difference between the fraction of trials during which the rats went to the side ipsilateral to side of its optical implant for inactivation and control trials for each of 10 binned stimulus strengths. Thus, a positive value resulting from this measurement represents an increase in ipsilateral responses in laser illumination trials over control trials in which the optical stimulation was absent. Confidence intervals and statistical comparisons for this bias parameter were calculated using nonparametric bootstrap procedures. The bias resulting from unilateral pharmacological inactivation was calculated in a similar way, but the control behavior was derived from non-inactivation control sessions obtained the day before inactivation. The performance impairment resulting from bilateral pharmacological inactivation sessions was also calculated using the non-inactivation control sessions obtained the day before inactivation. Performance was defined as percent correct trials for each binned stimulus strength.

## Analysis of neural recording

Spike waveforms were sorted on the basis of their relative energies and amplitudes on different channels of each tetrode. Clustering software (SpikeSort3D, Neuralynx) was used to isolate single units manually. Each spike was graphically positioned in a two- or three-dimensional space representing the energy or amplitude of the spike on two or three of the four tetrode channels. Convex hull boundaries and template-matching of waveforms were used to identify well-separated clusters of spikes, which were individually color coded. Data from the entire session were spike-sorted together. To compute the peri-event time histogram (PETH) for the population activity in response to the presentation of auditory clicks (*Figure 3a* and *Figure 3—figure supplement 1*), we followed the following procedure. For all well-isolated single units, individual trial rate functions were first

generated by smoothing the spike trains with a causal half-Gaussian filter with 0.1 s standard deviation. The response functions of individual neurons were then normalized on the basis of the mean firing of each individual neuron at the time of stimulus onset. Trials were subsequently sorted by a quantity that we defined as the 'mean stimulus strength' following the same procedure that has been described previously (*Hanks et al., 2015*). Mean stimulus strength was defined by dividing trials for each neuron into quantiles that are based on a difference in the preferred and non-preferred click rates.

The influence of single auditory clicks on neural responses, the 'click-triggered average', was calculated as follows. The trials of individual neurons were first grouped on the basis of the underlying Poisson rates that were used to generate the auditory stimuli. For each group, the mean PETH was computed. This quantity corresponds to the expected neural response at each point in time for each Poisson rate group. This mean response was then subtracted from each trial to generate the residual response from the expected one given the Poisson rate. Aligning this residual response to a click describes the change in the neural response that is associated with a single auditory click relative to the average expected response to clicks at other times. These click-aligned residual responses were averaged across all click times to obtain the click-triggered average response for each Poisson rate group. The click-triggered average for each neuron was calculated by averaging across the different Poisson rate groups. To compute the response across clicks arriving from both the preferred and the non-preferred sides, we inverted the residual response for non-preferred direction clicks prior to averaging. The click-triggered average response profiles generated using this procedure were compared to a model-based prediction that was based on a graded, linear encoding of accumulated evidence. To do this, we simulated evidence accumulation trajectories for 5000 trials using the same range of stimulus difficulties and durations that existed for the neural data. We then encoded these simulated trajectories with a graded, linear function of accumulated evidence (firing rate $r = k_1 \times a(t) + k_2$ in which $k_1$ and $k_2$ are constants). Finally, we applied the same analysis described for the neural data to estimate the predicted click-triggered average under this encoding (*Figure 3b*).

## Behavioral model-based analysis of neural data

We applied recently developed methods in our lab to generate tuning curves that specify the relationship between neural firing rates and mentally accumulated evidence (*Hanks et al., 2015*). These techniques take advantage of a behavioral model that provides a moment-by-moment and trial-by-trial estimation of the mentally accumulated evidence for this task (*Brunton et al., 2013*; *Hanks et al., 2015*).

### Neural tuning curves

Following *Hanks et al. (2015)*, the behavior model's estimate *a(t)* was related directly to neural firing rates on individual trials to estimate neural tuning curves for accumulating evidence. The estimates of the neural response and accumulating evidence in individual trials were used to calculate the joint probability distribution between those two variables as a function of time for each neuron. The correspondence between time in the model and neural time was determined on the basis of the latency of the stimulus-dependent response modulation. This latency was calculated as the first time bin in the PETH to have a significant modulation of neural response based on stimulus strength, which corresponded to 150 ms (*Figure 3a*). Thus, t = 0 in the model was taken as 150 ms after stimulus onset. From the joint probability, we extracted each neuron's response conditional on the value of the accumulator. We then combined across neurons by weighting the contribution of each by the inverse of the variance of this conditional distribution, which gives more weight to representations that are less noisy.

To quantify the relationship between neural response and accumulator value across time, we averaged across the time period from 0.15 to 0.5 s into the decision process. To characterize the encoding across individual neurons, we fit this relationship of the response to the accumulator value with a four-parameter sigmoid using the following equation:

$$r = k_1 + \frac{k_2}{1 + e^{-k_3(a-k_4)}}$$

In this equation, $k_2 k_3/4$ determines the slope at zero-crossing, which characterizes whether the

neural response changes smoothly between negative and positive accumulator values or whether it changes sharply in this region.

## Additional information

### Funding

| Funder | Grant reference number | Author |
|---|---|---|
| National Institutes of Health | R01MH108358 | Carlos D Brody |
| Starr Foundation | Starr Fellowship | Michael M Yartsev |
| National Institutes of Health | F32MH098572 | Timothy D Hanks |

The funders had no role in study design, data collection and interpretation, or the decision to submit the work for publication.

### Author contributions

Michael M Yartsev, Timothy D Hanks, Conceptualization, Resources, Data curation, Software, Formal analysis, Supervision, Funding acquisition, Validation, Investigation, Visualization, Methodology, Writing—original draft, Project administration, Writing—review and editing; Alice Misun Yoon, Software, Formal analysis, Visualization, Writing—original draft; Carlos D Brody, Conceptualization, Supervision, Funding acquisition, Writing—review and editing

### Author ORCIDs

Michael M Yartsev (iD) http://orcid.org/0000-0003-0952-2801
Timothy D Hanks (iD) http://orcid.org/0000-0003-4147-4475
Alice Misun Yoon (iD) http://orcid.org/0000-0001-7832-2796
Carlos D Brody (iD) https://orcid.org/0000-0002-4201-561X

### Ethics

Animal experimentation: All animal procedures described in this study were approved by the Princeton University Institutional Animal Care and Use Committee (IACUC; Protocols #1853) and carried out in accordance with National Institutes of Health standards.

### Decision letter and Author response

Decision letter https://doi.org/10.7554/eLife.34929.026
Author response https://doi.org/10.7554/eLife.34929.027

## Additional files

### Supplementary files

• Supplementary file 1. Best-fit parameters. This table shows the values of the parameters that maximize the likelihood of the eight-parameter accumulator model for all no-perturbation sessions of each rat (four rats for pharmacological inactivation study, three rats for electrophysiology, and 13 rats for optogenetics). The 95% confidence range is shown in brackets underneath each estimate.
DOI: https://doi.org/10.7554/eLife.34929.017

• Supplementary file 2. Best fit parameters (unilateral striatum inactivation data, n = 4 rats). Table 2 shows the values of the parameters that maximize the likelihood of the 11-parameter accumulator model for unilateral striatum inactivation data (n = 3528 trials) and unilateral striatum control data (n = 8349 trials). The 95% confidence range is shown in brackets underneath each estimate. The control corresponds to data taken on the no-infusion days immediately prior to the infusion day. Infusion sessions were spaced apart from each other by at least, and often more, non-infusion days.
DOI: https://doi.org/10.7554/eLife.34929.018

• Supplementary file 3. Best fit parameters (bilateral striatum inactivation data, n = 3 rats). Table3 shows the values of the parameters which maximize the likelihood of the 8-parameter accumulator

model for the bilateral striatum inactivation data (n = 2266 trials), the fit to the bilateral striatum control data (n = 3846 trials). 95% confidence range is shown in brackets underneath each estimate. Control corresponds to data taken on the no-infusion days immediately prior to the infusion day. Infusion sessions were spaced apart from each other by at least seven, and often more, non-infusion days.
DOI: https://doi.org/10.7554/eLife.34929.019

• Supplementary file 4. Best fit parameters (unilateral FOF inactivation data, n = 10 rats). Table 4 shows the values of the parameters that maximize the likelihood of the 11-parameter accumulator model for the unilateral FOF inactivation data (n = 3836 trials), and the fit to the unilateral FOF control data (n = 47,580, trials). The 95% confidence range is shown in brackets underneath each estimate. The control corresponds to data taken on the no-infusion days immediately prior to the infusion day. Infusion sessions were spaced apart from each other by at least seven, and often more, non-infusion days.
DOI: https://doi.org/10.7554/eLife.34929.020

• Supplementary file 5. Best fit parameters (unilateral striatum inactivation data, n = 4 rats). Table 5 shows the values of the parameters that maximize the likelihood of the eight-parameter accumulator model for the unilateral striatum inactivation data (n = 1809 trials) and bilateral FOF control data (n = 1531 trials). The 95% confidence range is shown in brackets underneath each estimate. The control corresponds to data taken on the no-infusion days immediately prior to the infusion day. Infusion sessions were spaced apart from each other by at least seven, and often more, non-infusion days.
DOI: https://doi.org/10.7554/eLife.34929.021

• Supplementary file 6. BIC – 11 p full model (unilateral inactivation). Table 6 shows, for each parameter, the difference in Bayesian Information Criterion (BIC; *Nagin, 1999*) for the 11-parameter full model, minus the BIC for the model with that parameter removed. Positive numbers thus favor keeping the parameter. Calculations are for data for unilateral striatum inactivations (n = 3528 trials) and the corresponding control data (n = 8349 trials).
DOI: https://doi.org/10.7554/eLife.34929.022

• Supplementary file 7. BIC – 8 p full model (bilateral inactivation). Table 7 uses the same conventions and shows, for each parameter, the difference in BICs for the eight-parameter full model minus a model with that parameter removed. Calculations are for data from bilateral striatum inactivations (n = 2266 trials) and for corresponding control data (n = 3846 trials).
DOI: https://doi.org/10.7554/eLife.34929.023

• Transparent reporting form
DOI: https://doi.org/10.7554/eLife.34929.024

## Data availability

All data generated or analysed during this study are included in the manuscript and supporting files.

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
