## [Decision Letter]

Thank you for submitting your article "Causal contribution and dynamical encoding in the striatum during evidence accumulation" for consideration by *eLife*. Your article has been reviewed by three peer reviewers, including Joshua Gold as the Reviewing Editor and Reviewer #1, and the evaluation has been overseen by Timothy Behrens as the Senior Editor. The following individual involved in review of your submission has agreed to reveal their identity: Mark D Humphries (Reviewer #2).

The reviewers have discussed the reviews with one another and the Reviewing Editor has drafted this decision to help you prepare a revised submission.

Summary:

This paper combines multiple strands of evidence to support the hypothesis that the anterior dorsal striatum (ADS) plays an important and causal in evidence accumulation in the brain. Pharmacological inactivation results in a strong impairment of decision-making. Catch trials for lesioned animals show decision impairment is independent of a motor deficit. Fits to evidence-accumulation models suggest three independent effects of ADS inactivation: 1) a shortening of the evidence integration window; 2) a dramatic increase in sensory and/or integrator noise; and 3) increased random choices on the contra-lateral side (lapse rate). Individual unit activity showed ramp-like increases on average over trials that matched key model predictions describing a continuous accumulation of evidence. Finally, acute optogenetic inactivation using eNpHR3 reproduced the lesion results, but only during the evidence integration epoch of the task, strongly suggesting a causative role for the ADS in evidence integration.

The reviewers agreed that this is a strong paper that addresses an interesting and timely topic and presents several compelling lines of evidence. However, the reviewers also agreed that there were a number of serious concerns that should be addressed, listed below.

Essential revisions:

1) The manuscript, especially the Introduction, should do a much better and more accurate job of crediting prior studies, and clarifying the specific, new contributions from this study. For example, the first work using the clicks task should be cited (Sanders and Kepecs). Likewise, Ding and Gold, 2012 used a perturbation of neural activity in the striatum to test for a causal role in evidence accumulation but is mentioned only and incorrectly as showing "correlates of evidence accumulation" in terms of "firing rates." In addition, the claim that the evidence for evidence accumulation in striatum comes only from trial averages of spike rates is false: Ding (2015, "Distinct dynamics of ramping activity in the frontal cortex and caudate nucleus in monkeys") used analyses that go beyond simple spike rates to identify correlates of evidence accumulation.

2) There were a number of questions about the behavioral model fits and their interpretation. Several of these questions centered on the estimation and interpretation of lapse rates. The group data in Figure 1B,C seem to show relatively high (~20%) lapses, but the values tended to be much lower in Table 1. This discrepancy would seem to imply that other parameters of the behavioral model can account for errors for the strongest stimuli. Is that the case? Which parameters? Are such effects justified; that is, is it reasonable to think that errors for those stimuli result from inefficiencies in the decision process? If so, can lapses be effectively estimated using the given range of stimulus values?

More generally, the reviewers appreciated the descriptions of the model-fitting procedures but were not fully convinced that the use and interpretation of such a complex model (in terms of number of parameters) is justified. Could reduced and/or alternative models provide more parsimonious accounts of the data, given penalties for model complexity? Are all of the parameters needed? Do confidence intervals on their best-fitting values lend insights into the reliability of their estimates and therefore interpretation? How do the parameters trade-off? For example, an input-gain bias for one side could be plausibly offset by a post-categorization bias to the other, leading to no net effect. There are hints of this in Table 1, where the "control" fits show wide variation in some parameters between animals. A lack of consistent fits between animals would then lead to potentially spurious divisions into "significant" and "non significant" differences between two groups of animals, simply depending on how the parameter values tended to cluster or not cluster by chance. Furthermore, what evidence supports the claim that the "combination of sensory plus accumulator noise" represents a single, interpretable quantity, and that it has explanatory power beyond the effects of muscimol on lapse rates?

Given these sources of uncertainty about the model fits, how justified is the conclusion that ADS inactivation affects the accumulation process, as opposed to, say, a choice bias in the presence of uncertainty and/or lapse rates? It might be useful to use fits and/or simulations of models that include only biases and/or lapses to determine if and how well they can reproduce the experimental effects. These issues apply to both the pharmacological and optogenetic manipulations.

3) Please comment on apparent discrepancies between Figure 1B, C and 2A,C. Why the larger range of stimulus values on the abscissa in Figure 2? Why are the effects on ipsilateral choices in Figure 1B not apparent in Figure 2A?

4) Why do the analyses in Figure 4 only cover up to 0.5 s after stimulus onset? Ding and Gold, 2010 reported correlates of evidence accumulation early, but not late, in the stimulus presentation period. What happens to the late spiking activity here?

5) The evidence presented here for the role of ADS in evidence accumulation does not satisfy the criterion of "necessary": decision-making was impaired, not prevented. The claims in the paper should be adjusted accordingly.

---

## [Author Response]

Essential revisions:1) The manuscript, especially the Introduction, should do a much better and more accurate job of crediting prior studies, and clarifying the specific, new contributions from this study. For example, the first work using the clicks task should be cited (Sanders and Kepecs). Likewise, Ding and Gold, 2012 used a perturbation of neural activity in the striatum to test for a causal role in evidence accumulation but is mentioned only and incorrectly as showing "correlates of evidence accumulation" in terms of "firing rates." In addition, the claim that the evidence for evidence accumulation in striatum comes only from trial averages of spike rates is false: Ding (2015, "Distinct dynamics of ramping activity in the frontal cortex and caudate nucleus in monkeys") used analyses that go beyond simple spike rates to identify correlates of evidence accumulation.

We are very grateful to the reviewers for giving us the opportunity, and their help, for improving the Introduction. We are particularly embarrassed at having failed to cite Dong and Gold’s, 2012 perturbations. Together with the same authors’ 2010 neural recordings, those perturbations were a major part of the motivation for our own work, and we were both dismayed and embarrassed to realize we had failed to cite them.

We’re also embarrassed to have failed to cite Sanders and Kepecs, 2012. We have completely rewritten the Introduction, to more specifically describe the background literature and its various components (including citing Sanders and Kepecs, 2012, Ding and Gold, 2012, and Ding, 2015), and to describe the motivation for our work and better describe what, given the context in the literature, is new in the current manuscript.

2) There were a number of questions about the behavioral model fits and their interpretation. Several of these questions centered on the estimation and interpretation of lapse rates. The group data in Figure 1B,C seem to show relatively high (~20%) lapses, but the values tended to be much lower in Table 1. This discrepancy would seem to imply that other parameters of the behavioral model can account for errors for the strongest stimuli. Is that the case? Which parameters? Are such effects justified; that is, is it reasonable to think that errors for those stimuli result from inefficiencies in the decision process? If so, can lapses be effectively estimated using the given range of stimulus values?More generally, the reviewers appreciated the descriptions of the model-fitting procedures but were not fully convinced that the use and interpretation of such a complex model (in terms of number of parameters) is justified. Could reduced and/or alternative models provide more parsimonious accounts of the data, given penalties for model complexity? Are all of the parameters needed? Do confidence intervals on their best-fitting values lend insights into the reliability of their estimates and therefore interpretation? How do the parameters trade-off? For example, an input-gain bias for one side could be plausibly offset by a post-categorization bias to the other, leading to no net effect. There are hints of this in Table 1, where the "control" fits show wide variation in some parameters between animals. A lack of consistent fits between animals would then lead to potentially spurious divisions into "significant" and "non significant" differences between two groups of animals, simply depending on how the parameter values tended to cluster or not cluster by chance. Furthermore, what evidence supports the claim that the "combination of sensory plus accumulator noise" represents a single, interpretable quantity, and that it has explanatory power beyond the effects of muscimol on lapse rates?Given these sources of uncertainty about the model fits, how justified is the conclusion that ADS inactivation affects the accumulation process, as opposed to, say, a choice bias in the presence of uncertainty and/or lapse rates? It might be useful to use fits and/or simulations of models that include only biases and/or lapses to determine if and how well they can reproduce the experimental effects. These issues apply to both the pharmacological and optogenetic manipulations.

This point 2 from the reviewers contained multiple interesting questions and concerns. We have organized our responses immediately below in a slightly different order to how the questions were posed, but we have made sure to cover all of the questions raised in point 2.

2.1 […] Generally, the reviewers appreciated the descriptions of the model-fitting procedures but were not fully convinced that the use and interpretation of such a complex model (in terms of number of parameters) is justified. […] Could reduced and/or alternative models provide more parsimonious accounts of the data, given penalties for model complexity? Are all of the parameters needed?

Following the request from reviewers to take into account penalties for model complexity, and to further quantify how each parameter is needed, we have added Supplementary file 6 and 7, which shows, for each parameter, the difference in Bayesian Information Criterion (BIC) between a reduced model with that parameter deleted, minus the full model. Negative numbers indicate that the BIC favors the reduced model with the parameter deleted, while positive numbers indicate favoring keeping the full model. We found that except for the accumulator bounds parameter B, each parameter has one or more conditions that indicates we should keep the parameter.

We also point out that each of the parameters describes a distinct effect, and each of the parameters is significantly different from zero, i.e., it has a statistically significant effect on the log likelihood (Brunton et al., 2013). With one exception, they are thus all needed to fully describe the data. The exception is once again, consistent with the BIC measures, the sticky commitment bound “B”, which across every individual rat in this manuscript (but not in other papers) had a confidence interval that included infinity (see parameter fit tables in supplementary files; B = infinity corresponds to not having the B parameter at all). We have nevertheless chosen, for consistency across papers, to keep that parameter. It is possible that future perturbation studies, in other brain regions, might reveal an effect on B, in which case it will be important to have it documented here that B was not affected by perturbations of the ADS; for that reason, we have kept B.

In sum, except for the accumulator bounds parameter B, which we keep for consistency with other papers, both BIC and confidence intervals indicate we should keep all parameters.

2.2 How do the parameters trade-off? For example, an input-gain bias for one side could be plausibly offset by a post-categorization bias to the other, leading to no net effect.

Thank you for this question, this is a point that we had touched on in previous work, but had done so only buried in a Supplementary Figure (see Supplementary Figure S4 in Brunton, 2013), so it is very helpful to clarify it here. The short answer is that parameter trade-offs are made explicit, and quantified, by the shape of the region in parameter space that fits the data well, i.e., the region of high likelihood, as shown in Figure 2.

A succinct description of the shape of the peak of the log likelihood is given by the covariance matrix of a Gaussian that is fit around the optimum (an approach known as the Laplace approximation (MacKay, 2003)). We have added a section to the Materials and methods titled “Trade-offs between different model parameters” that goes through this procedure.

One such parameter trade-off that we found in our pharmacological inactivation data, referred to explicitly in the paper, is between sensory and accumulator noise parameters (Figure 2D and related text).

In addition, the reviewers’ question regarding how input-gain bias to one side might be offset by post-categorization bias to the other side led us to realize that we should clarify that many (but not all) potential parameter trade-offs are in fact dispelled by use of a model that utilizes knowledge of the individual click times. We have therefore added the following to the main text before Figure 2:

“We note that although below, for simplicity of presentation, we visualize some of the results of the model fits in terms of psychometric plots (i.e., graphing the probability of a decision to one side as a function of total #R – #L clicks, averaged over trials), the model and its fits are sensitive to the detailed timing of the click stimuli in each individual trial, information that is obscured in the trial-averaged psychometric plot. […] The shape of the ridge quantifies the extent and scaling of the trade-off (Materials and methods and e.g. Figure 2D in Brunton et al., 2013).”

2.3. There are hints of [parameter trade-offs] in Table 1, where the "control" fits show wide variation in some parameters between animals. A lack of consistent fits between animals would then lead to potentially spurious divisions into "significant" and "non significant" differences between two groups of animals, simply depending on how the parameter values tended to cluster or not cluster by chance.

We apologize for the confusion, we should have clarified that our statistical significance on perturbations is assessed on data combined across rats. We have added the following to the first paragraph of the “Behavioral Model” section of the Materials and methods:

“As in previous work (Erlich et al., 2015), to obtain sufficient number of perturbation trials for the model-based analysis, we combined data across rats and assessed effects and statistical significance on this “meta-rat” data. For completeness, in Supplementary file 1 we also present the result of fits to each individual rat, which are in general consistent with the meta-rat conclusions.”

2.4. Do confidence intervals on their best-fitting values lend insights into the reliability of their estimates and therefore interpretation?

Yes, absolutely, we should have clarified that confidence intervals are examined throughout. We now clarify in the caption for Figure 2 that the likelihood landscape panels shown there are intended specifically to show confidence intervals, or more precisely, confidence *regions* (“region” rather than “interval” since the model has multiple parameters). The likelihood landscapes indicate both the parameter trade-offs and the confidence regions: the shape of the confidence region describes the trade-offs because it defines the region in parameter space within which the likelihood is high and within which the parameters can trade off with each other.

To make sure that this point does not go unnoticed, we have added an arrow to the color scale in Figures 2B,D to explicitly indicate that the 95% confidence region limits are defined in the panels shown.

In sum, the confidence regions in Figure 2 and (Figure 2—figure supplement 1) are there precisely to quantify the reliability of the parameter estimates, and therefore their interpretation, which includes how parameters trade off with each other.

2.5. Several of [the] questions [about the behavioral model fits and their interpretation] centered on the estimation and interpretation of lapse rates. The group data in Figure 1B,C seem to show relatively high (~20%) lapses, but the values tended to be much lower in Table 1. This discrepancy would seem to imply that other parameters of the behavioral model can account for errors for the strongest stimuli. Is that the case? Which parameters?

Yes, that is the case, and the noise parameters largely account for the errors at strongest stimuli.

We have added Figure 2—figure supplement 3, showing the same models, with the same parameters, but now run on simulations with the noise parameters set to zero (which would produce high total signal-to-noise ratios). In these cases, the model-predicted asymptotic performance now matches the lapse rate parameter.

In other words, as the reviewers indicated, this means that performance in the trials of Figure 1 has not reached the asymptotic best level allowed by the lapse rate in the model parameters. Consequently, other parameters must be limiting performance for the easiest trials shown in Figure 1, and these are the noise parameters.

2.6. Are such effects justified; that is, is it reasonable to think that errors for those stimuli result from inefficiencies in the decision process? If so, can lapses be effectively estimated using the given range of stimulus values?

We believe these effects are indeed justified and that lapses can be effectively estimated, for the following reason: Lapse rates are defined as probabilities of error independent of any stimulus characteristics (number of clicks, difference in number clicks, inter-click intervals, etc.). Defined as such, they can be distinguished from other parameters, even when using only a limited range of stimulus values, because other parameters, including the noise parameters, do affect behavior in a way that depends, and therefore varies with, stimulus characteristics. Performance limits that remain after accounting for stimulus-characteristic dependent limits are the lapse rate.

As above, multidimensional confidence regions quantify parameter trade-offs, thus showing which parameters can be distinguished from each other given our data, and indicate the reliability of parameter estimates. See response *2.7* immediately below for distinguishing lapse rates from noise parameters.

2.7. [What evidence supports the claim that the "combination of sensory plus accumulator noise"] has explanatory power beyond the effects of muscimol on lapse rates?

In the new Figure 2—figure supplement 1 we show a number of panels examining parameter trade-offs in our data, including a panel in ED Figure 5A (third row, middle column) that shows that in fact the lapse rate κ_C_ does not trade off with the summed sensory plus accumulator noise. This lack of a trade-off with lapse rates is also true for individual noise parameters, although we did not add a panel for each one. The two types of parameters (lapse versus noise) have different explanatory power, and are distinguished (using our behavioral model that takes into account each individual click time) because, as described above, lapse rates and the different noise parameters have different dependencies on stimulus characteristics. We now allude to this in the text, where we added:

“Lapse rate and noise parameters described distinct effects, and did not trade off with each other (Figure 2—figure supplement 1A, third row, middle column)”

2.8. Furthermore, what evidence supports the claim that the "combination of sensory plus accumulator noise" represents a single, interpretable quantity?

We apologize for the confusion, we did not mean to imply that the combination of sensory plus accumulator noise is a single biologically-interpretable quantity. Instead, we simply mean that those parameters trade off in such a way that our data does cannot distinguish between multiple different combinations of them. To make this clear, we have added the following to the text:

“We note that we do not mean to imply that the combination of sensory and accumulator noise is a single, biologically-interpretable quantity, but simply that our data cannot distinguish between the different trade-offs between these that fit the data equally well.”

For example, in Figure 2D a whole range of trade-offs can be seen. E.g., all three of (σ^2^_a_ = 130, σ^2^_s_ = 25) and (σ^2^_a_ = 90, σ^2^_s_ = 100) and (σ^2^_a_ = 70, σ^2^_s_ = 150) fit the data well (see also Figure 2—figure supplement 1).

2.9. Given these sources of uncertainty about the model fits, how justified is the conclusion that ADS inactivation affects the accumulation process, as opposed to, say, a choice bias in the presence of uncertainty and/or lapse rates? It might be useful to use fits and/or simulations of models that include only biases and/or lapses to determine if and how well they can reproduce the experimental effects. These issues apply to both the pharmacological and optogenetic manipulations.

We apologize for the confusion, we did not mean to imply that the modeling results conclusively show that the ADS is involved in the accumulation process. The modeling results do suggest that, are consistent with that idea, and furthermore suggest the more specifically intriguing possibility that the ADS may be, in some as-yet-to-be-determined way, particularly important for controlling accumulator noise levels. But we think of the modeling results, by themselves, as suggestive, not conclusive. To clarify our view of the modeling results, we have added the following at the top of the paragraph in the discussion related to modeling:

“By identifying model parameters affected by the inactivations, our model fits suggest specific aspects of the evidence accumulation computation that could be prioritized as potentially particularly strongly related to the ADS’s role in the computation.”

And we have modified the following sentence in that discussion paragraph:

“Nevertheless, even while we emphasize that we do not take the modeling results on their own as conclusive, they do suggest accumulator noise as a principal parameter of interest.”

The evidence that we *do* think strongly points to the ADS being involved in the accumulation process come from the three main questions: (1) is this region necessary for unimpaired accumulation of evidence? (2) Does this region encode graded accumulating evidence? (3) Is this region causally involved through the accumulation time period? These are the three main questions that are study answers affirmatively, and that we raise in the Abstract and in the Introduction.

3) Please comment on apparent discrepancies between Figure 1B, C and 2A,C. Why the larger range of stimulus values on the abscissa in Figure 2? Why are the effects on ipsilateral choices in Figure 1B not apparent in Figure 2A?

We apologize for the apparent discrepancy. This was due largely to different methods of binning the data. Figure 1 used bins with matched numbers of trials per bin, while Figure 2 used bins with equal spacing along the abscissa. Due to the higher density of trials with low click differences, that entailed a wider range along the abscissa for the latter method. We have adopted the former method for both plots for consistency.

4) Why do the analyses in Figure 4 cover only up to 0.5 s after stimulus onset? Ding and Gold, 2010 reported correlates of evidence accumulation early, but not late, in the stimulus presentation period. What happens to the late spiking activity here?

The analyses cover this time period due to our experimental design of using an exponential distribution for determining the stimulus duration. This leads to a larger percentage of short duration trials, and fewer trials of longer duration. This contrasts with the reaction time design of Ding and Gold, 2010, where monkeys typically responded with RTs greater than 0.5 s for even the fastest conditions and for over 1 s for more difficult, slower conditions. We agree this is an important consideration, but we do not have a clear idea of what would happen at longer durations. It is entirely conceivable that our results would align with Ding and Gold’s for these longer durations. We have revised the discussion to raise this important consideration.

5) The evidence presented here for the role of ADS in evidence accumulation does not satisfy the criterion of "necessary": decision-making was impaired, not prevented. The claims in the paper should be adjusted accordingly.

We agree with the reviewers that the statement of necessity should be adjusted. Throughout the paper we have replaces “necessary” to now more precisely say that the ADS “is required for unimpaired performance.”